# Tetrameric architecture of an active phenol-bound form of the AAA$^+$ transcriptional regulator DmpR

Kwang-Hyun Park [1,6], Sungchul Kim [2,6], Su-Jin Lee[1,3], Jee-Eun Cho[1], Vinod Vikas Patil [1,3], Arti Baban Dumbrepatil [1], Hyung-Nam Song[1], Woo-Chan Ahn[1], Chirlmin Joo [2✉], Seung-Goo Lee [4], Victoria Shingler[5] & Eui-Jeon Woo [1,3✉]

The *Pseudomonas putida* phenol-responsive regulator DmpR is a bacterial enhancer binding protein (bEBP) from the AAA$^+$ ATPase family. Even though it was discovered more than two decades ago and has been widely used for aromatic hydrocarbon sensing, the activation mechanism of DmpR has remained elusive. Here, we show that phenol-bound DmpR forms a tetramer composed of two head-to-head dimers in a head-to-tail arrangement. The DmpR-phenol complex exhibits altered conformations within the C-termini of the sensory domains and shows an asymmetric orientation and angle in its coiled-coil linkers. The structural changes within the phenol binding sites and the downstream ATPase domains suggest that the effector binding signal is propagated through the coiled-coil helixes. The tetrameric DmpR-phenol complex interacts with the $\sigma^{54}$ subunit of RNA polymerase in presence of an ATP analogue, indicating that DmpR-like bEBPs tetramers utilize a mechanistic mode distinct from that of hexameric AAA$^+$ ATPases to activate $\sigma^{54}$-dependent transcription.

[1] Disease Target Structure Research Center, Korea Research Institute of Bioscience & Biotechnology (KRIBB), Daejeon 305-806, Republic of Korea. [2] Kavli Institute of Nanoscience and Department of Bionanoscience, Delft University of Technology, 2629 HZ Delft, The Netherlands. [3] Department of Proteome Structural Biology, KRIBB School of Bioscience, University of Science and Technology (UST), Daejeon 305-333, Republic of Korea. [4] Synthetic Biology and Bioengineering Research Center, Korea Research Institute of Bioscience & Biotechnology (KRIBB), Daejeon 305-806, Republic of Korea. [5] Department of Molecular Biology, Umeå University, 90187 Umeå, SE, Sweden. [6] These authors contributed equally: Kwang-Hyun Park, Sungchul Kim. ✉email: C.Joo@tudelft.nl; ejwoo@kribb.re.kr

The AAA$^+$ family of ATPases is involved in various essential cellular processes. The bacterial enhancer binding (bEBP) subgroup of AAA$^+$ proteins couple ATPase hydrolysis to initiation of transcription by $\sigma^{54}$-RNA polymerase ($\sigma^{54}$-RNAP)[1]. Many bEBPs belong to two-component systems, in which a membrane-bound histidine kinase senses and transfers a signal from the environment to a corresponding response regulator to allow $\sigma^{54}$-dependent promoter activity[2]. In contrast, some bEBPs are single-component sensory regulators that directly bind effector molecules to achieve the same outcome[3]. DmpR (di-methyl phenol regulator) from *Pseudomonas putida* KCTC 1452 (also known as CapR) is a single-component bEBP that serves as a sensor of phenolic compounds[4–6]. In habitats contaminated by phenol and other aromatic pollutants, catabolism of these compounds is mediated by tightly regulated operons that encode specialized suites of enzymes necessary for the sequential breakdown of recalcitrant compounds (e.g., toluene, xylene, cresols and other aromatic ring-containing hydrocarbons)[7]. DmpR has also been widely used in engineering of bacteria and the development of whole-cell biosensors[8–10].

As is typical of bEBPs, DmpR consists of three domains: (1) a sensory domain consisting of a vinyl-4-reductase (V4R) scaffold that functions in binding of an aromatic effector molecule[11–13], (2) a conserved central AAA$^+$ ATPase domain bearing the bEBP-specific GAFTGA motif that is involved in coupling ATP hydrolysis to the restructuring of $\sigma^{54}$-RNAP, and (3) a DNA binding domain that interacts with the palindromic upstream activating sites (UASs) situated ~100–200 bp upstream from the $\sigma^{54}$ promoter[14]. The B-linker that connects the sensory domain and the ATPase domain plays an important role in relaying the effector binding signal to allow ATP hydrolysis[15]. In hexameric bEBPs with ring structures, higher-ordered oligomers induce formation of the catalytic active site at the interface between adjacent ATPase subunits[16]. DmpR share high sequence homology with other aromatic-responsive bEBPs, such as XylR, TouR, PoxR and MopR, and this subgroup are known to transition from inactive dimers to active oligomers upon the binding of an aromatic effector compound as a prerequisite for their capacity to direct $\sigma^{54}$-dependent transcription[1,17].

Although the ATPase domains of bEBPs generally mediates oligomerization into the active multimeric form, the internal signal transduction mechanism that results in oligomerization upon aromatic effector binding is not yet fully understood. In particular, the exact number of subunits within the active oligomer and how they are arranged to enable a productive interaction with $\sigma^{54}$-RNAP has remained unknown for more than two decades. Similarly, the mechanism underlying negative regulation mediated by the sensory domain—so that truncates lacking this domain exhibit aromatic effector-independent transcriptional promoting activity—has likewise not been fully explained[18]. Here, we determined the oligomerization status of DmpR by a single-molecule fluorescence imaging technique, present a tetrameric structure of the phenol-bound DmpR complex and demonstrate its capacity to interact with $\sigma^{54}$. As the report of a tetrameric bEBP capable of interacting with $\sigma^{54}$, the conformational change observed in the DmpR-phenol complex provides a structural basis for understanding the signal transduction activation mechanism of DmpR-like single-component bEBPs.

purity >95%; 66 kDa). As assessed by blue native (BN)-PAGE analysis, in the absence of phenol, DmpR$^{WT}$ appeared as a mixture of dimers (~132 kDa) and tetramers (~264 kDa). When incubated with 1 mM phenol, the band corresponding to the dimer shifted to reflect the higher molecular weight of the DmpR$^{WT}$ tetramer (Supplementary Fig. 1a). A change in the oligomeric sate of DmpR$^{WT}$ by phenol was also observed in size exclusion chromatography (SEC) and in dynamic light scattering (DLS), respectively (Supplementary Fig. 1b, c). Addition of ATP analogues (ANP-PNP or ATP$\gamma$S), or DNA containing its specific binding sites (upstream activating sequences, UASs) did not change the tetrameric association of DmpR$^{WT}$ (Supplementary Fig. 1d). DmpR$^{WT}$ exhibited a marginal increase in DNA binding activity in the presence of phenol ($K_D$ value ~ 387 nM) as compared to the absence of phenol (~476 nM) (Supplementary Fig. 1e). Consistent with these findings, multi-angle light scattering (MALS) analysis also showed a protein peak with a molecular weight of ~280 kDa upon the addition of phenol in both the presence and the absence of ATP$\gamma$S, indicating that DmpR$^{WT}$ predominantly forms a tetramer in response to phenol (Supplementary Fig. 1f). The presence of a tetrameric sub-population before addition of phenol presumably resulted from binding of *E. coli* derived aromatic metabolites as has been observed for some other aromatic hydrocarbon binding proteins[11,12,20,21].

To confirm tetramer formation upon phenol binding, we used single-molecule photobleaching (SMPB)[22,23]. We generated a fusion containing fluorescent eGFP and N-terminally 6 × His-tagged DmpR$^{WT}$ (Fig. 1a). The fusion protein was surface-immobilized using a biotinylated anti-GFP antibody. Stepwise bleaching signals from eGFP were recorded using total internal reflection fluorescence (TIRF) microscopy (Fig. 1b). A TIRF image of eGFP-DmpR$^{WT}$ showed clearly separate fluorescent spots (Fig. 1c). Discrete steps were observed from individual eGFP-DmpR$^{WT}$ fluorescence time traces (Fig. 1d). Although there were ~18% of protein aggregates (Fig. 1e), eGFP-DmpR$^{WT}$ exhibited a photobleaching distribution that corresponds to a mixture of multiple oligomeric states. In the absence of phenol, a major fraction of molecules (~32%) showed two-step photobleaching, which is indicative of dimers. One-step (monomers), three-step (trimers) and four-step (tetramers) photobleachings were observed in around 11, 14 and 19% of the population, respectively (Fig. 1f). One-step bleaching from dimeric eGFP-DmpR$^{WT}$ and less-than-four-step bleaching from tetrameric eGFP- DmpR$^{WT}$ could originate from incomplete eGFP maturation[22,24,25]. The eGFP maturation was estimated to be 85% from the ratio between a protein concentration measured from the 280-nm absorbance and an eGFP fluorophore concentration measured from 488-nm absorbance. There were hardly any oligomers that underwent more than five photobleaching steps within the populations. Upon the addition of phenol, a majority of the molecules (~34%) exhibited four-step bleaching, while ~17% of the molecules exhibited two-step bleaching, indicating that phenol promotes an increase of the tetrameric population at the expense of the dimer population (Fig. 1g, i, j). No change was observed upon the addition of ATP (Fig. 1h–j). Together, these results show that phenol promotes tetramer formation by DmpR$^{WT}$ and this oligomerization is independent of ATP.

## Results

**Phenol promotes tetrameric association.** Upon the addition of a phenolic ligand, DmpR forms oligomers which are required to promote transcription[19]. We first examined the formation of oligomers in response to the addition of phenol using purified full-length DmpR bearing an N-terminal 6 × His tag (DmpR$^{WT}$,

**Structure of the tetrameric DmpR$^{\Delta D}$-phenol complex.** Purification and crystallization of DmpR$^{WT}$ was hampered by a limited amount of full-length protein due to its low solubility and aggregation as inclusion bodies in *E. coli*. Based on a solubility profile analysis and preliminary tests, we designed a truncated

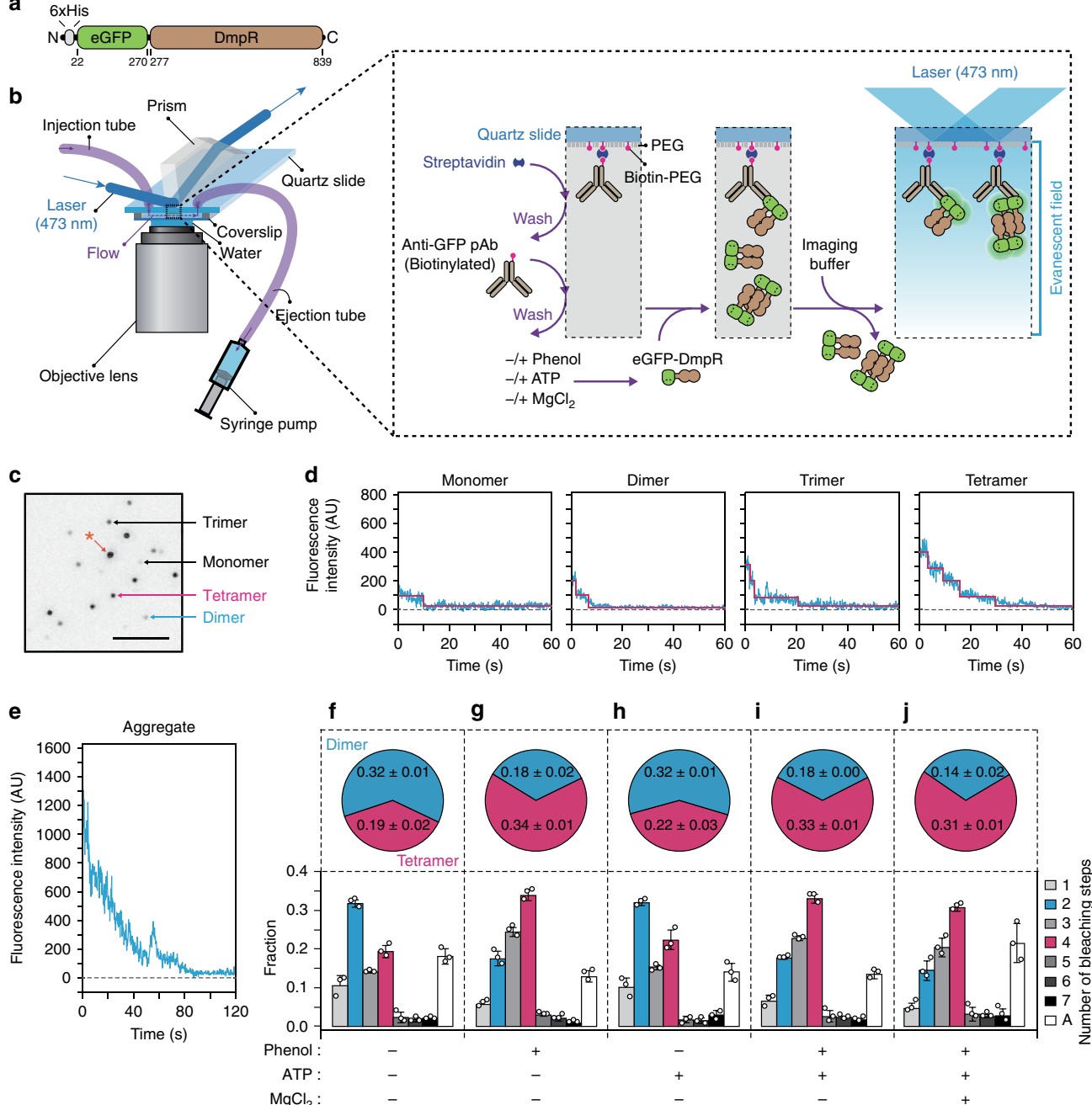

**Fig. 1 Single-molecule stoichiometry measurements of DmpR oligomers. a** Domain organisation of the eGFP-DmpR protein used for single-molecule photobleaching (SMPB). **b** Schematic overview of the experimental design of SMPB assays. **c** A representative EMCCD image including four major species of eGFP-DmpR proteins (monomer, dimer, trimer and tetramer). Asterisks represent the signal from presumable protein aggregates. All data are representative of five replicates with similar results. Scale bars, 5 μm. **d** Representative time trajectories of the eGFP emission signals. The stoichiometry of the eGFP-DmpR proteins was determined by counting the number of eGFP photobleaching steps. Light blue lines are eGFP emission traces. Pink lines represent stepwise fits of the traces. **e** A representative time trajectory of the signal from presumable protein aggregates. **f–j** Distribution of photobleaching steps of eGFP-DmpR. The pie graphs above the histograms depict the ratio of dimers and tetramers for each condition. Events with more than eight photobleaching steps were categorised as aggregates (A). Data are presented as mean ± SD from three independent experimental replicates with $n \geq 180$ individual molecule (Counts).

DmpR derivative (aa 18–481) that is soluble and produced at sufficient levels in *E. coli*. This truncated protein, DmpR$^{\Delta D}$, has an N-terminal $6 \times$ His tag that replaces the first 15 residues, lacks the DNA binding domain, and carries serine substitutions of two cysteine residues (C119S/C137S) that were anticipated to be located at the protein surface. DmpR$^{\Delta D}$ has a phenol binding affinity

($K_D = \sim 12$ μM) similar to full-length DmpR ($K_D = \sim 16$ μM)[19] and likewise exhibits tetrameric oligomerization in the presence of phenol (Supplementary Fig. 2a, b).

The determined crystal structure of DmpR$^{\Delta D}$ shows a phenol molecule bound to the sensory domain of each protomer. The sensory and ATPase domains are connected by an ~35 Å long

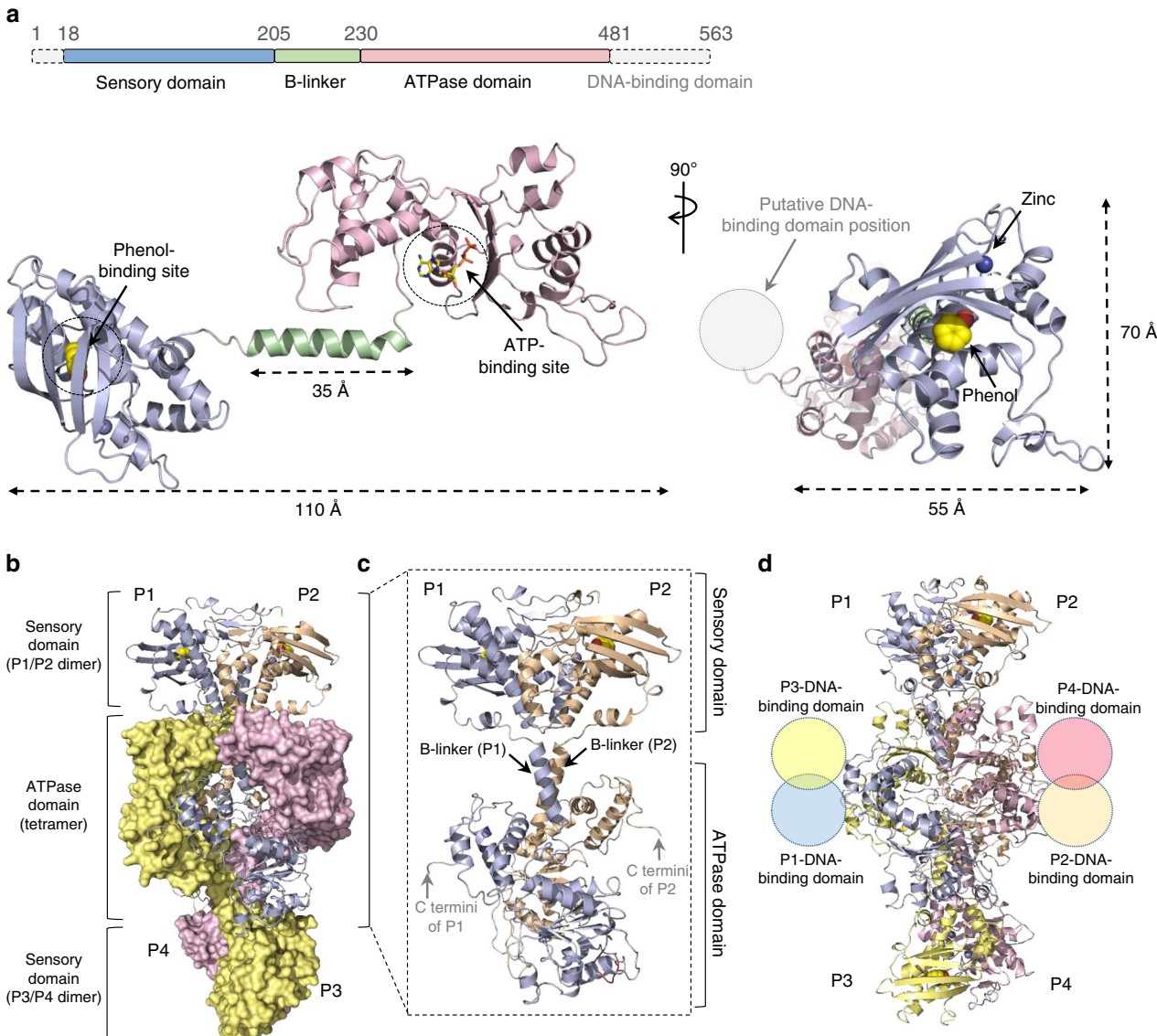

**Fig. 2 Overall structure of the DmpR^ΔD-phenol complex. a** Upper: domain organisation of the DmpR^ΔD protein. Lower: protomer structure of phenol- and zinc-bound DmpR^ΔD with side (left) and top-down (right) views. The putative position of the DNA-binding domain is indicated by a grey dotted circle. **b** Overall structure of the tetrameric DmpR^ΔD-phenol complex. Each protomer is individually represented, with the P1 (light blue)/P2 (wheat) dimer shown as cartoons and the P3 (pale yellow)/P4 (light pink) dimer shown as surface representations. **c** Structure of the P1/P2 head-to-head dimeric DmpR^ΔD-phenol complex. **d** The putative location of the DNA-binding domains within the tetrameric complex.

helical B-linker. The protomer topology exhibits a 'dumbbell-like' structure with approximate dimensions of $110 \times 55 \times 70$ Å (Fig. 2a). The DmpR^ΔD-phenol complex is a dimer-of-dimers with overall dimensions of $150 \times 75 \times 70$ Å (Fig. 2b). The two protomers (P1 and P2) form an elongated intertwined P1/P2 dimer through extensive interactions between the related sensory domains and parallel coiled-coil B-linkers in a head-to-head orientation with a buried surface area of 2895 Å² (Fig. 2c). The two dimers—P1/P2 and P3/P4—form the tetramer, which has an antiparallel head-to-tail assembly that places the four ATPase domains at the central core of the complex. The complex, with dimeric sensory domains at either end, adopts an overall elliptical rod-like shape. Since the two DmpR^ΔD C-termini are located next to each other due to the twofold symmetry, the DNA binding domains that are missing in truncated DmpR^ΔD would be present as pairs, and those from the P1 and P3 protomers would be on one side and those from the P2 and P4 protomers would be on

the opposite side of the centre of the complex, as depicted in Fig. 2d.

The formation of the DmpR^ΔD-phenol complex buries a surface area of ~26,800 Å² (33% of the combined surfaces) between the protomers. The P1/P2 sensory domain dimer packs against the ATPase domains in the P3/P4 dimer in such a way that the Val53 and Ile58 residues in the P1 sensory domain interact with Phe312 in the GAFTGA motif (aa 310–315) in the P3 ATPase domain (Supplementary Fig. 3a). Residues Glu210 and Glu214 in the P1 protomer B-linker interact with Thr316 and Arg319 of the GAFTGA loop within the ATPase domain of the P4 protomer (Supplementary Fig. 3b). The same pattern is observed for the P2 and P3 protomers. The pairs of ATPase domains within the P1/P2 and P3/P4 dimers do not interact with each other (Supplementary Fig. 3c), whereas the ATPase domains in the P1 and P2 protomers interact with those in the P3 and P4 protomers, respectively, through the twofold symmetry observed

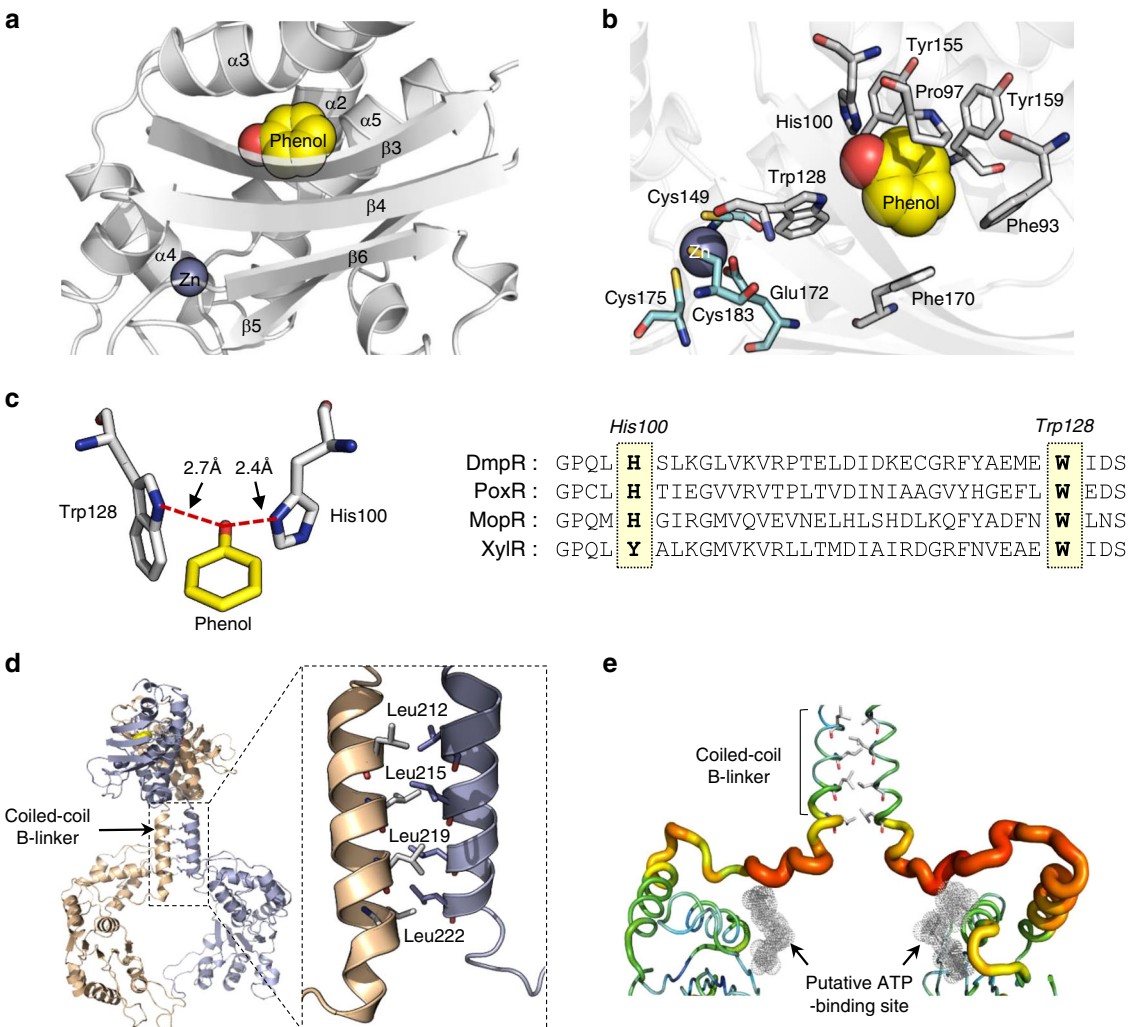

**Fig. 3 Phenol-binding pocket and the coiled-coil B-linkers. a** Positions of phenol and zinc inside the core $(\beta/\alpha)_4$ barrel scaffold of the sensory domain. **b** Key residues in the phenol-binding pocket and residues involved in zinc coordination are shown as sticks. **c** Depiction of the hydroxyl group of phenol interacting with His100 and Trp128 (left); sequence alignment of five aromatic compound-binding transcriptional activators (right). DmpR, *P. putida* KCTC 1452; PoxR, *Ralstonia sp.* E2; MopR, *Acinetobacter guillouiae*; and XylR, *Pseudomonas putida*. **d** The interface of the coiled-coil B-linkers with the strips of leucine residues highlighted within the enlarged box. **e** B-factor putty tube representation of the B-linker in the ATPase domain connection loop. Orange and red colours and a wider tube indicate regions with high B-factors, whereas shades of green and narrower tubes indicate regions with low B-factors. The putative ATP-binding site is shown in dot representation.

between the α-helical P1/P3 and P2/P4 subdomains (Supplementary Fig. 3d).

**Phenol-bound sensory domain and B-linker.** The sensory domain of DmpR shows a core $(\beta/\alpha)_4$ barrel scaffold with a bound phenol and zinc ion (Fig. 3a). Each N-terminal region, comprised of residues 18–45 in each sensory domain, intertwines with the other sensory domains to yield a tightly interlocked homodimer. The phenol is located in an enclosed cavity (24–36 $\text{Å}^3$ in volume) formed by an antiparallel hairpin motif. The cavity is primarily lined by hydrophobic residues, including Phe93, Trp128, Tyr155, Tyr170, and Tyr159. A strictly conserved Trp128 residue is located between the phenol-binding site and the zinc-binding site, while the zinc is coordinated by residues Cys151, Glu172, Cys177 and Cys185 (Fig. 3b). The hydroxyl group of the phenol is located between His100 and Trp128, indicating a ligand-positioning function of these residues. His100 is conserved in other phenol-responsive regulatory proteins, such as PoxR and MopR, while it is substituted by tyrosine in the toluene/xylene-responsive XylR (Fig. 3c). Interestingly, the electron density of the

phenol group is strong in the P1 protomer, whereas it is weak in the P2 protomer (Supplementary Fig. 4a, b). The same pattern is also observed in the P3/P4 dimer. Given its location inside a closed pocket, the weak electron density suggests low occupation by phenol in the binding cavities of the P2 and P4 protomers, which is associated with the altered conformations of the two protomers and the asymmetric shape of the B-linkers (see below).

The B-linker connects each lobe of the sensory and ATPase domains to form a linear helix with leucine residues at positions 212, 215, 219 and 222 creating a hydrophobic stripe on one side of the helix in the amphipathic structure. These strips of leucine residues in the two B-linkers adopt a coiled-coil architecture in the dimer and exhibit knobs-into-holes packing typical of leucine zippers (Fig. 3d)[26]. At the end of the B-linker, the helix connects to a flexible loop region consisting of residues 227–240 that has a high B-factor (~27 Å) and a sharp angle that extends into the ATPase domain (Fig. 3e).

**The ATPase domain and its tetramer-dependent activity.** The ATPase domain consists of a typical α/β subdomain (aa 236–401)

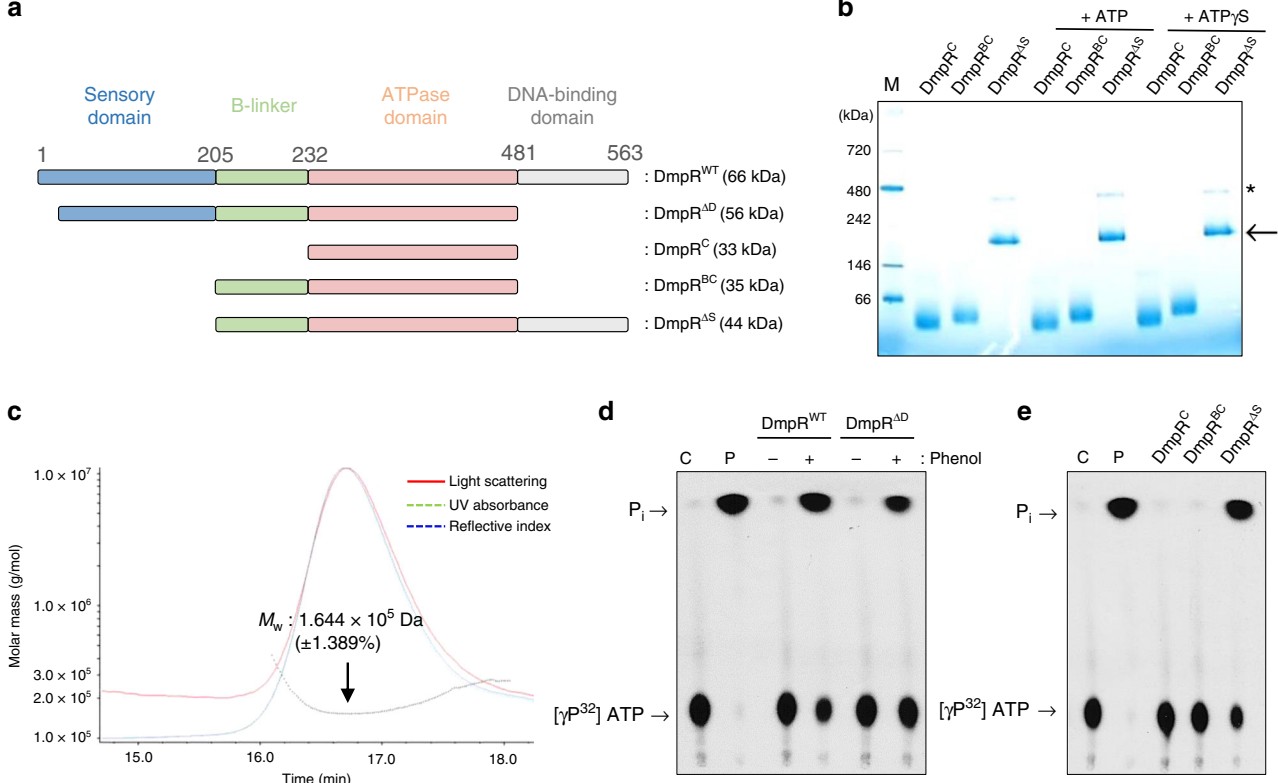

**Fig. 4 Oligomerization and ATPase activities of DmpR derivatives. a** Schematic diagram of wild type DmpR (DmpR$^{WT}$) and truncated DmpR proteins (DmpR$^{\Delta D}$, residues 18–481; DmpR$^C$, residues 232–481; DmpR$^{BC}$, residues 205–481; and DmpR$^{\Delta S}$, residues 205–563). The molecular weights given include the N-terminal 6 × His tag. **b** BN-PAGE of truncated DmpR proteins after the addition of ATP or ATPγS. Bands corresponding to tetramers are marked by an arrow. Asterisks indicate an artefact band of DmpR$^{\Delta S}$. This data is representative of three replicates with similar results. **c** SEC combined with multi-angle light scattering (MALS) analysis to calculate the molecular weight (MW) of DmpR$^{\Delta S}$ (dotted black line). **d** ATPase activity of DmpR$^{WT}$ and DmpR$^{\Delta D}$ in the presence or absence of phenol. Hydrolysis of [γ-P$^{32}$] ATP to generate P$^{32}$ was visualised by thin-layer chromatography. 'C' represents the reaction with the reaction buffer as control. 'P' represents the reaction with the alkaline phosphatase as positive control. This data is representative of five replicates with similar results. **e** ATPase activities of truncated DmpR derivatives. This data is representative of five replicates with similar results.

and an α-helical subdomain (aa 402–481). The GAFTGA motif (aa 310–315) of the P1 ATPase domain is located close to the P3 sensory domain (aa 53–59) and the P4 B-linker helix (aa 209–213) (Supplementary Fig. 5a). The GAFTGA regions exhibit conformational variation among the DmpR protomers, indicating their flexibility (Supplementary Fig. 5b). All of the ATPase domains have an overall structure similar to that of the ADP-bound form of PspF[27]. Although the crystallization of DmpR$^{\Delta D}$ occurred in the presence of 3 mM AMP-PNP, no electron density corresponding to a nucleotide was observed, suggesting that the GAFTGA conformations in this structure may reflect an inactive state that is poised to bind ATP. The putative ATP binding site (cavity volume of ~26 Å$^3$) lies at the interface between the α/β subdomain and the α-helical subdomain and is spatially placed so that residues Glu232 and Tyr233 from the flexible loop that connects the B-linker and the ATPase domain could potentially interact with an ATP molecule (Supplementary Fig. 5c)[28]. Arg223, which is conserved in the B-linkers of aromatic-sensing DmpR-like bEBPs, is located in the proximity of the putative ATP binding site of the adjacent protomer (Supplementary Fig. 5d).

To investigate the connection between oligomerization and ATPase activity, we purified additional truncated derivatives of DmpR (Fig. 4a). BN-PAGE analysis of these derivatives after incubation with ATP or ATPγS revealed that both the ATPase domain alone (DmpR$^C$) and the ATPase domain attached to the B-linker (DmpR$^{BC}$) exhibited a monomeric conformation. However, the truncated protein lacking only the sensory domain (DmpR$^{\Delta S}$)

displayed a tetrameric conformation even in the absence of phenol (Fig. 4b). Similarly, SEC-MALS analysis showed a peak corresponding to a protein with a molecular weight of ~164 kDa, indicating that DmpR$^{\Delta S}$ predominantly forms tetramers in solution (Fig. 4c). The trace band of higher molecular weight observed in BN-PAGE in DmpR$^{\Delta S}$, but not in DmpR$^C$, DmpR$^{BC}$ or DmpR$^{WT}$, is likely an artefact caused by non-native self-interaction of the sensory domain deleted DmpR protein. These results show that the ATPase domain of DmpR$^C$ alone, or when attached to the B-linker (DmpR$^{BC}$), does not multimerize despite the major contribution of the ATPase domain to tetramer formation by DmpR$^{\Delta D}$. These findings additionally suggest the involvement of the DNA-binding domain in tetramer formation, possibly through the pairing of the DNA-binding domains[29]. Next, we investigated the ATPase activity of all the DmpR derivatives to assess the correlation between oligomerization and ATPase activity. The DmpR$^{WT}$ and DmpR$^{\Delta D}$ proteins exhibited ATP hydrolysis in the presence of phenol, but they exhibited only marginal ATP hydrolysis in the absence of phenol (Fig. 4d). In contrast, the monomeric DmpR$^C$ and DmpR$^{BC}$ derivatives did not show any ATP hydrolysis activity, while the tetrameric DmpR$^{\Delta S}$ protein exhibited efficient ATPase activity irrespective of the addition of phenol (Fig. 4e). These results suggest that a tetrameric configuration is essential and sufficient for the ATPase domains of DmpR to hydrolyse ATP.

**Alteration of the conformations within an asymmetric shape.** Conformational changes of DmpR were revealed when the

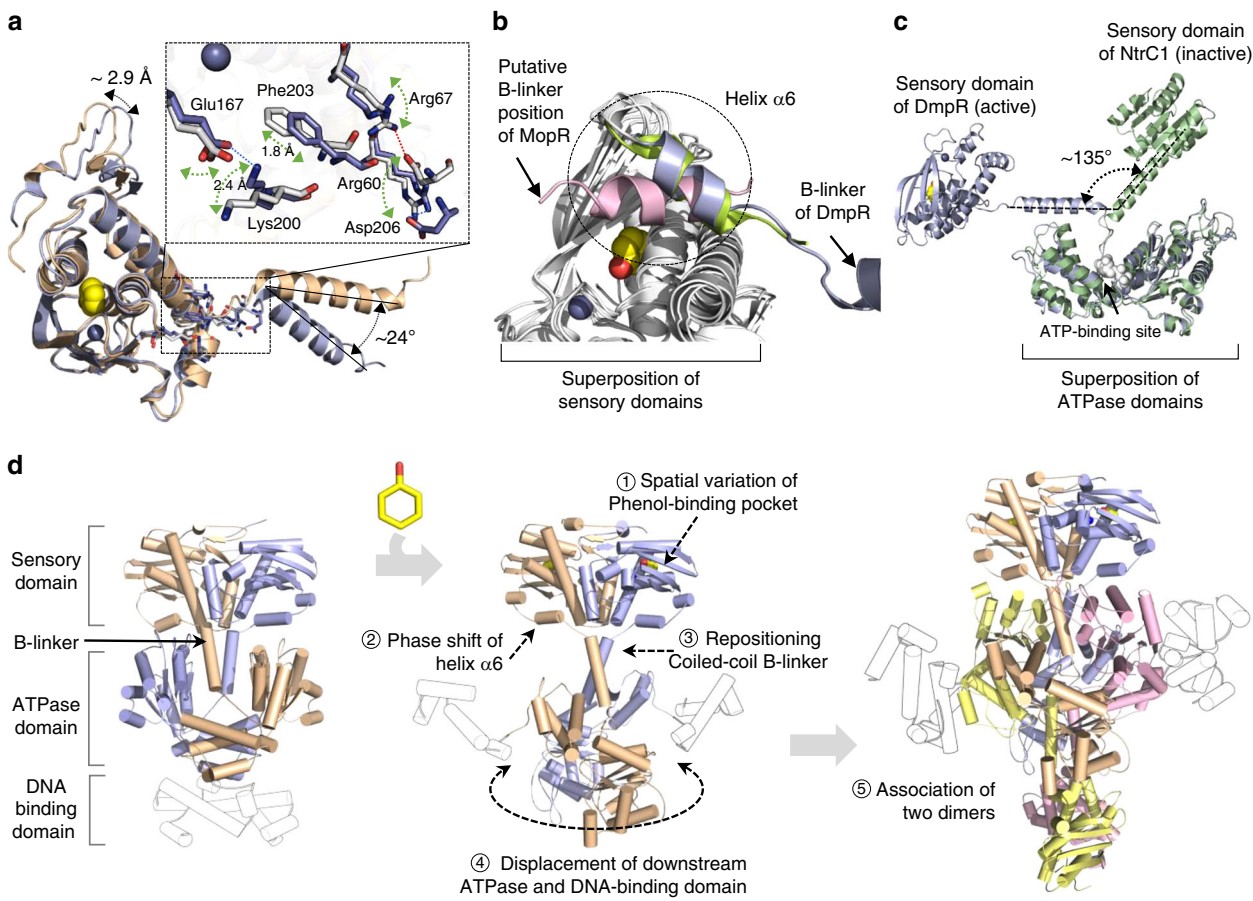

**Fig. 5 Conformational changes within DmpR$^{\Delta D}$. a** Superimposition of the sensory domains of the P1 protomer (light blue) and the P2 protomer (wheat) to highlight the asymmetry. The region showing significant structural changes is indicated by a dotted box. The movement of several residues in the region connecting helix α6 and the B-linker (blue sticks in the P1 protomer and white sticks in the P2 protomer). The changes in each residue are indicated by the dotted arrows. **b** Structural flexibility of helix α6 in the sensory domains of MopR (PDB ID, 5kbi, light pink), PoxR (PDB ID, 5fru, yellow) and DmpR (light blue) is represented within a dotted circle. **c** Superimposition of the DmpR protomer (light blue) onto that of NtrC1 (PDB ID, 1ny5, green) with respect to the ATPase domains to highlight the flexible region and the different trajectories of the B-linker and ATPase domain. **d** Model of the conformational change of the transition from the inactive DmpR dimer to the tetrameric complex upon the binding of phenol.

protomer structures were overlapped. Superimposition of the P1 protomer, which has a high phenol occupancy, onto that of the P2 protomer, which has a low phenol occupancy, uncovered interesting structural features. The volume of the phenol-binding pocket in the P1 protomer was 23.59 Å$^3$, whereas it was 36.91 Å$^3$ in the P2 protomer due to marginal shifts in the residues lining the pocket, including Tyr90, Phe93, His100, Val113, Phe122, Tyr159 and Phe170 (Supplementary Fig. 6a). The N-terminal region, which is involved in the interlocking of dimers (aa 18–39), is located ~2.9 Å further away from the phenol-binding site in the P1 protomer than in the P2 protomer. The helices in the B-linker also differ, with those from the P2 protomer adopting an orientation off-set by ~24° compared to that of the corresponding helix from the P1 protomer, and as a result, the dimer exhibits a notably asymmetric configuration (Fig. 5a). The same pattern of conformational variation was observed in the P3/P4 dimer across the diagonal of the complex (P1/P2, r.s.m.d. = 3.7 Å, 437Cα; P1/P3, r.s.m.d. = 0.9 Å, 443Cα; and P1/P4, r.s.m.d. = 4.0 Å, 443Cα).

The significant shift in the B-linker is associated with helix α6 in the sensory domain; Lys200 is involved in a charged interaction with Glu167 in the P1 protomer at a distance of 2.4 Å, while Phe203 is shifted 1.8 Å further away from the sensory domain in the P1 protomer than in the P2 protomer. Asp206 from the P1 protomer is involved in a charged interaction with

Arg60 in the sensory domain, whereas the same residue in the P2 protomer points outside of the helix and is closer to Arg67 (Fig. 5a). Interestingly, the position of the α6 helix exhibits significant variation among the DmpR, PoxR and MopR structures despite the high structural similarity in other regions of the sensory domain (PoxR, r.s.m.d. = 1.1 Å, 196Cα; MopR, r.s.m.d. = 0.9 Å, 158Cα) (Supplementary Fig. 6b)[11,12]. Helix α6 in MopR shows a completely opposite trajectory to that observed in DmpR, demonstrating the flexibility of this helical region among the subfamily members (Fig. 5b).

The closest structural analogue of the DmpR monomer is NtrC1, which is a bEBP member of a two-component system (Z score = 28.9, r.s.m.d. = 1.7 Å for 247 Cα). Superimposition of the ATPase domains of DmpR$^{\Delta D}$ with those of inactive NtrC1 (PDB ID, 1ny5) highlights the significantly altered orientation of their B-linkers. With respect to the ATPase domain, the cognate B-linkers are displaced by ~135° despite the high structural similarity of each module (B-linker, r.s.m.d. = 1.3 Å, 21 Cα; ATPase domain, r.s.m.d. = 2.4 Å, 243 Cα) (Fig. 5c). A recent report showed that the central AAA$^+$ domain and part of the B-linker of apo DmpR forms a homodimer with a face-to-face orientation in the ATPase domain[28]. Given the head-to-head geometry of the tightly intertwined sensory domains of DmpR and the dimeric features of the coiled coil B-linker helixes, the apo dimer of DmpR may adopt a configuration similar to that of the

inactive dimer of NtrC1 or NtrX (Supplementary Fig. 6c)[30,31]. Overall, the spatial variation in the phenol binding pocket, the phase shifts of the residue interactions in helix α6 and the asymmetric angle and trajectory of the B-linker of DmpR indicate propagation of structural changes and modulation of downstream domain interactions through the B-linker upon phenol binding (Fig. 5d) (see below).

**Interaction between tetrameric DmpR and σ54.** Activation of transcription involves a physical interaction between the bEBP and σ54-RNAP, specifically through the N-terminus region (aa 1–56) of the σ-factor[32]. We examined the interaction of the ligand-bound DmpR complex with σ54 using far-Western blotting[33] (Supplementary Fig. 7a). A band corresponding to the size of the σ54 protein was detected only when DmpR$^{WT}$ was incubated in the presence of phenol and ATPγS (Supplementary Fig. 7b), while the ATPase activity of DmpR$^{WT}$ did not change upon addition of the σ54 protein (Supplementary Fig. 7c). We next measured the interaction of DmpR$^{WT}$ with the σ-factor using isothermal titration calorimetry (ITC) with σ54$_{(1–119)}$-CPD. The σ54$_{(1–119)}$-CPD protein comprises the N-terminal residues of σ54 (aa 1–119) fused to a C-terminal cysteine protease domain (CPD) that allowed better expression and purification (Supplementary Fig. 7d). Consistent with the far-Western data, DmpR$^{WT}$ interacted with the N-terminal peptide of σ54 only in the presence of phenol and ATPγS ($K_D$ = ~4 µM; Supplementary Fig. 7e). The stoichiometry of the ITC binding curve ($n$ = 0.86 ± 0.022) indicates a 1:1 molar ratio for the interaction between σ54$_{(1–119)}$-CPD and tetrameric DmpR$^{WT}$.

To visualise the interaction of DmpR with σ54$_{(1–119)}$-CPD and confirm the stoichiometry of the complex, we used single-molecule fluorescence imaging[34]. In the first series of experiments, biotinylated σ54$_{(1–119)}$-CPD was surface-immobilized through a biotin-streptavidin interaction, and then stepwise photobleaching signals from σ54$_{(1–119)}$-CPD bound eGFP-DmpR$^{WT}$ were recorded using TIRF microscopy (Fig. 6a and Supplementary Fig. 8a). The number of binding events between eGFP-DmpR$^{WT}$ and σ54$_{(1–119)}$ significantly increased upon the addition of phenol and ATPγS (Fig. 6b and Supplementary Fig. 8b). As assessed by real-time imaging, the majority of eGFP-DmpR$^{WT}$ bound to σ54$_{(1–119)}$ exhibited four-step bleaching under these conditions (Fig. 6c, d and Supplementary Fig. 8c). The fractions of the monomeric, dimeric and trimeric states could be attributed to the incomplete maturation of the eGFP fluorophore[22,24,25,35]. These results show that when associated with phenol and ATPγS, tetrameric DmpR efficiently interacts with the σ54 peptide.

As a complementary approach, we reversed the order of the interaction by immobilizing eGFP-DmpR$^{WT}$ in the presence of phenol and ATPγS using biotinylated anti-GFP antibody. Cy5-labelled σ54$_{(1–119)}$-CPD was then added to assess the interaction between DmpR and the σ54$_{(1–119)}$ peptide and determine which oligomeric state(s) of DmpR can interact with σ54 (Fig. 6e and Supplementary Fig. 8d, e). Binding of Cy5-labelled σ54$_{(1–119)}$-CPD co-localized with surface-immobilized eGFP-DmpR, indicating a highly specific interaction. eGFP-DmpR$^{WT}$ binding, which was observed at a location where σ54$_{(1–119)}$ was pre-bound (Fig. 6f), further revealed that tetrameric DmpR specifically interacts with the σ54$_{(1–119)}$ peptide (Fig. 6g). Taken together, the single-molecule data suggests that in the presence of phenol and ATP, tetrameric DmpR binds σ54 to activate transcription by σ54-RNAP.

## Discussion

Research on the activation mechanism of DmpR has been hindered due to the ambiguity of the oligomeric state of its

transcription-promoting active form. DmpR has been widely believed to form hexamers[13], primarily based on its similarity to ring-structured hexameric bEBPs such as NtrC and PspF[36,37]. Although many AAA$^+$ ATPases function as hexamers, the active oligomeric state of DmpR-like bEBPs remained unclear. Thus, the discovery of the tetrameric configuration of DmpR and its demonstrated ability to interact with the σ54 factor provided by this study represents an important step for an increased understanding of the activation mechanism of DmpR-like single component bEBPs. Interestingly, the GAFTGA motif loops in the ATPase domains are located some distance from one another in the tetrameric architecture of DmpR with a perpendicular two-fold symmetry, whereas the GAFTGA loops are close together in the centre of the ring-like hexamer, indicating an altered mode of binding to σ54. The interaction of the DmpR tetramer with σ54 in a 1:1 ratio implies that the initial binding to σ54 likely occurs through a GAFTGA motif in a single ATPase domain. Such an interaction could plausibly cause a steric hindrance in the complex to prevent further interactions or could trigger an allosteric change in the tetramer that would allow it to assume a configuration optimally poised to activate σ54-RNAP; these two mechanistic alternatives require further investigation. Given the asymmetric configuration between two monomers in a dimer and the absence of ATP molecule in the crystal structure, the dynamic DmpR tetramer probably undergo conformational change in the process of binding and/or hydrolyzing ATP that accompanies its interaction with σ54.

The structural features of ligand-bound DmpR exhibit remarkable similarity to those of histidine kinases (HKs), which are sensory components of the bacterial two-component system. The sensing of environmental changes through a dimeric N-terminal domain, the shifts of the coiled-coil linker helixes in the middle of the molecule, and the modulation of ATPase activity by alterations in the positioning and orientation of a downstream domain are all reminiscent of the internal signal relay mechanism observed in HKs[2]. The coiled-coil architectures of the GAF, HAMP and PAS linker domains in HKs are known to be crucial for oligomerization, signalling and the regulation of their activity[38]. Although the exact mechanism of signal propagation through coiled-coil helixes in HKs is still under debate [e.g., an axial rotation, axial tilt (scissor) or axial shift (piston) mechanism], typically, HKs exhibit two distinct structural conformations: an "off" state that imposes conformational restraints on the downstream domains and a dynamic "on" state that releases those conformational restraints, allowing the downstream domains to carry out ATPase functions[39]. Intriguingly, the helical motifs that connect to the DHp domains in HKs reportedly exhibit asymmetric conformations[40], as observed in the DmpR$^{ΔD}$-phenol structure. Given that the symmetric to asymmetric "flip-flop" transition within a homodimer is a well-known signal transduction mechanism in many HKs[41–43], DmpR-like bEBPs may utilize a similar mechanism for signal transduction upon sensing aromatics. In particular, the formation of tetramers and the constitutive ATPase activity of the DmpR$^{ΔS}$ protein support the notion that the tightly-bound dimeric sensory domains of the full-length protein restrain the downstream domains to prevent tetramer formation in the absence of phenol, which explains the negative regulation of activity mediated by the sensory domain of DmpR. The tightly interlocked sensory domains, which are also observed in the PoxR and MopR structures[13,15], may also be the key structural element that would prevent hexamer formation and thus set DmpR and its homologues apart from the other typical hexameric AAA$^+$ ATPases.

In the absence of phenol, DmpR may form a dimer in such a way that the tightly intertwined sensory domains with a head-to-head geometry impose a conformational constraint on the coiled-

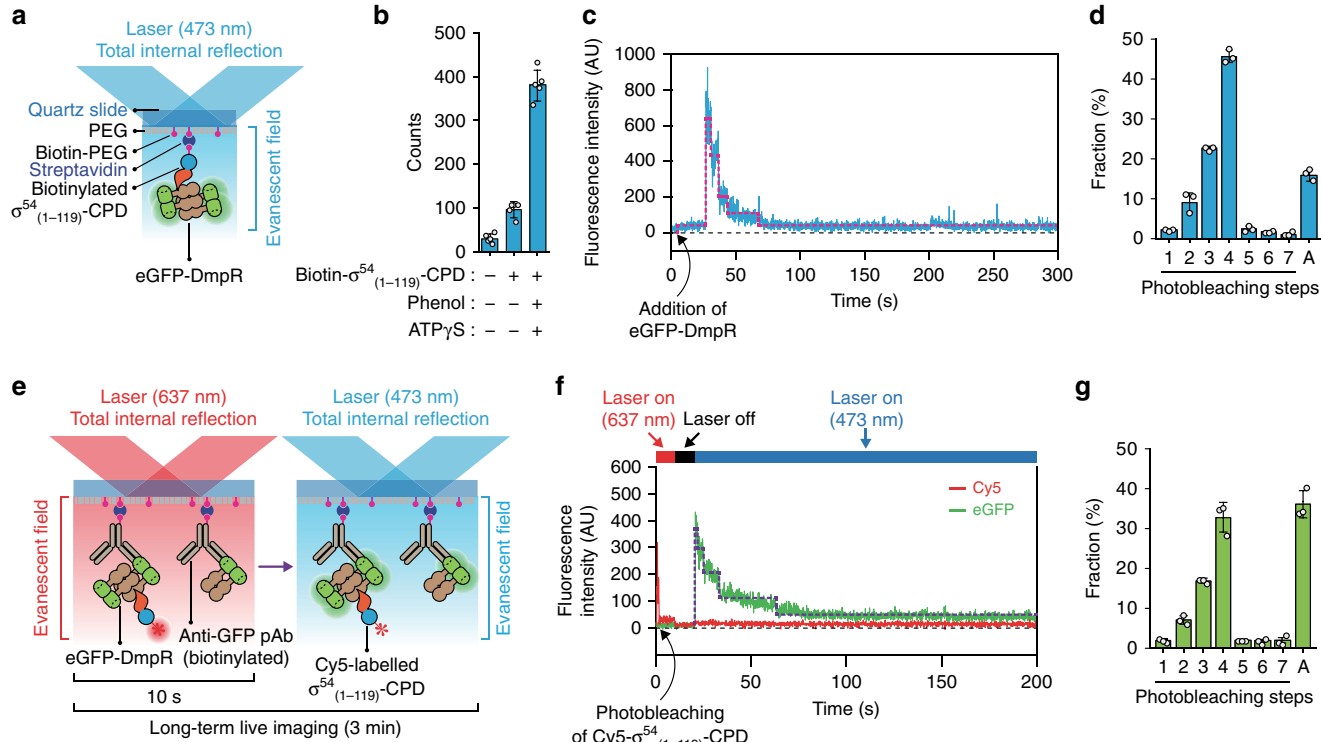

**Fig. 6 Single-molecule observation of the interactions of tetrameric DmpR with the N-terminus of σ54. a** Illustration of a single-molecule TIRF assay of eGFP-DmpR binding to σ54(1–119)-CPD. **b** The number of eGFP-DmpR complexes bound to σ54(1–119)-CPD without/with phenol and ATPγS per field of view (25 × 50 μm²). Data are presented as mean ± SD from five independent experimental replicates. **c** A representative time trace of tetrameric eGFP-DmpR binding to surface-immobilized σ54(1–119)-CPD. **d** Distribution of photobleaching steps of the eGFP-DmpR complexes bound to σ54(1–119)-CPD. Data represent the mean ± SD from three independent experimental replicates with n ≥ 980 individual molecule (Counts). Events with more than eight photobleaching steps were categorised as aggregates (A). **e** Schematic representation of a single-molecule TIRF assay of σ54(1–119)-CPD binding to surface-immobilized eGFP-DmpR. **f** A representative time trace of Cy5-labelled σ54(1–119)-CPD binding to antibody-tethered tetrameric eGFP-DmpR. Laser excitation (637 nm followed by 473 nm) is indicated above the trace. The photobleaching time point of the Cy5 signal is indicated. The photobleaching steps of eGFP-DmpR are represented by the dotted line (purple). **g** Distribution of photobleaching steps of the eGFP-DmpR complexes. eGFP signals co-localized with the signals of Cy5-labelled σ54(1–119)-CPD were analysed. Data represent the mean ± SD from three independent experimental replicates with n ≥ 185 individual molecule (Counts). Events with more than eight photobleaching steps were categorised as aggregates (A).

coil helixes to place the ATPase domains side-by-side. In this scenario, phenol binding in the sensory domain would induce a conformational change in the ligand binding pocket followed by the shift of the flexible α6 helix at the C-terminus of the sensory domain and a resultant change in the B-linker position. The rearrangement of the coiled-coil B-linkers within the dimer would alter the angles and interfaces of the downstream ATPase domains, allowing the association of two dimers in a head-to-tail orientation (Fig. 5d).

The observation of features common between the first (HK) and second (response regulator, e.g., NtrC) protein that make up two-component systems suggests that DmpR may have combined the sensing and the regulation modules of each protein into one protein to ensure simple and efficient detection of small lipophilic ligands that can freely diffuse through the membrane layer. The formation of a DmpR tetramer in the presence of phenol and the absence of ATP indicates that ATP binding and hydrolysis, known to be prerequisites for transcriptional activation, are not required for subunit association. It thus appears plausible that ATP is bound to DmpR after oligomerization, and the energy from ATP hydrolysis is subsequently utilized for coordinating the binding and restructuring of σ54-RNA polymerase through the structural rearrangement of the GAFTGA loop. Because it is structurally distinct from ring-forming hexameric AAA+ bEBPs, the interaction of a tetrameric complex with σ54 represents a

unique mechanistic mode of DmpR-like bEBPs in terms of σ54-dependent transcriptional activation.

## Methods

**Cloning and protein purification.** DNA encoding *DmpR* (Accession No. AAP46187.1) was amplified by PCR from *Pseudomonas putida* KCTC 1452 (Accession No. AF515710). Fragments spanning codons 1–563 (wild type), 18–481 (DmpR^ΔD), 205–563 (DmpR^ΔS), 205–481 (DmpR^BC) and 232–481 (DmpR^C) were cloned into the pProEX HTa vector, which has an N-terminal His tag (Invitrogen), via the *Bam*HI and *Hind*III restriction sites. The DmpR cysteine mutant (C119S/C137S) was generated using a site-directed mutagenesis kit (Enzynomics) and verified by DNA sequencing (Solgent). The σ54 gene sequence, (accession no. WP_003255133) including the σ54 and σ54(1–119) gene cassettes, was cloned into the pET22b expression vector via the *Nde*I/*Hind*III sites, and the CPD coding region was inserted in-frame using the *Hind*III/*Xho*I sites to generate the σ54(1–119)-CPD expression construct. The eGFP (FPbase ID. R9NL8) gene was fused with pProEX HTa-cloned DmpR by a ligation-independent cloning method. Detailed cloning primer information is listed in Supplementary Table 1.

For DmpR purification, the His-tagged wild type, substituted, and eGFP-tagged DmpR variants were produced using *E. coli* strain BL21-CodonPlus (DE3)-RIL (Agilent Technologies, #230245), which was cultured at 30 °C, with expression was induced with 1 mM IPTG. The cells were harvested, lysed and centrifuged. The supernatant was then applied to a His-Trap HP column (GE Healthcare) in elution buffer composed of 30 mM Tris-HCl (pH 7.5), 250 mM NaCl, 3 mM β-mercaptoethanol, 1 mM PMSF, 250 mM imidazole and 5% glycerol. The peak fractions were applied to a Superdex 200 Increase 10/300 column (GE Healthcare) in a final elution buffer composed of 30 mM Tris-HCl (pH 7.5), 50 mM NaCl, 3 mM β-mercaptoethanol and 5% glycerol. For σ54 purification, His-tagged wild type σ54 and its variants were expressed as described above. The supernatants were

applied to His-Trap HP columns (GE Healthcare) with elution buffer composed of 30 mM Tris-HCl (pH 7.5), 500 mM NaCl, 3 mM β-mercaptoethanol, 1 mM PMSF, 1 mM DTT, 250 mM imidazole and 5% glycerol. To remove the CPD tag, $\sigma^{54}$-CPD protein was incubated with 1 μM phytic acid overnight at 25 °C. Peak fractions were applied to a Superdex 200 Increase 10/300 column (GE Healthcare) in a final elution buffer composed of 30 mM Tris-HCl (pH 7.5), 150 mM NaCl, 3 mM β-mercaptoethanol and 5% glycerol.

To add the biotin and Cy5 fluorescent dye to $\sigma^{54}_{(1-119)}$-CPD, purified $\sigma^{54}_{(1-119)}$-CPD was reduced in phosphate-buffered saline (PBS) with 10 mM DTT for 2 hours at 25 °C. The reduced protein was buffer-exchanged into PBS without DTT using PD MiniTrap G-10 (GE Healthcare) and labelled with either poly(ethylene glycol) [N-(2-maleimidoethyl)carbamoyl]methyl ether 2-(biotinylamino)ethane (Sigma-Aldrich, cat# 757748) or Cy®5 Maleimide Mono-Reactive Dye (Sigma, cat# GEPA15131) for 2 hours at room temperature followed by incubation at 4 °C overnight. The labelled $\sigma^{54}_{(1-119)}$-CPD preparations were then purified by SEC with a Superdex 200 10/300 GL column. The fractions containing labelled proteins were concentrated using Amicon® Ultra Centrifugal Filters, pooled in PBS with 50% glycerol, snap-frozen in liquid nitrogen and stored at −80 °C.

**Single-molecule TIRF imaging and data acquisition**. A prism-type total internal reflection microscope was used for the SMPB experiments. The eGFP derivative was excited with a 473-nm laser (Coherent, OBIS LX 75 mW); Cy5 was excited using a 637-nm laser (Coherent, OBIS 637 nm LX 140 mW). To obtain time traces, eGFP was excited as weakly as possible to minimize their rapid photobleaching during the time course of a measurement. The fluorescence signals from single molecules were collected using an inverted microscope (Olympus, IX-73) with a ×60 water immersion objective (Olympus, ULSAPO60xW). To block the 473-nm laser scattering, we used a 473 nm EdgeBasic™ best-value long-pass edge filter (Semrock, BLP01-473R-25). When the 637-nm laser was used, the 637-nm laser scattering was blocked by a notch filter (Semrock, 488/532/635 nm, NF01-488/532/635). Subsequently, the Cy5 signals were spectrally split with a dichroic mirror (Chroma, 635dcxr) and imaged with the halves of an electron multiplying EMCCD camera (Andor Technology, iXon 897). The data were obtained in either single-colour or dual-colour mode.

To eliminate the nonspecific adsorption of proteins onto the quartz surface, piranha-etched slides (Finkenbeiner) were passivated with a mixture of mPEG-SVA (5 kDa, Lysan Bio Inc.) and Biotin-PEG-SVA (5 kDa, Lysan Bio Inc.) in the first PEGylation treatment, and then MS(PEG)4 Methyl-PEG-NHS-Ester reagent (ThermoFisher Scientific) was used for the second treatment as described previously[44]. To further improve the surface quality, the assembled microfluidic flow chambers were subsequently incubated with 5% Tween-20 (v/v in T50 buffer containing 10 mM Tris, pH 8.0, and 50 mM NaCl) for 10 min[45], followed by a wash step with 100 μL of T50 buffer. Afterwards, the slides were incubated with 50 μL of streptavidin (0.1 mg/mL in T50 buffer, S888, Invitrogen) for 5 min, followed by a wash step with 100 μL of phosphate-buffered saline (PBS).

For the single-molecule photobleaching (SMPB) assay, 50 μL of 1 ng/mL anti-GFP (biotin) goat polyclonal antibody (pAb) (Abcam, ab6658) was injected into the chambers and incubated for 5 min prior to a wash step with 100 μL of phosphate-buffered saline (PBS). One microlitre of 10 nM eGFP-DmpR^WT was incubated with or without 1 mM MgCl$_2$, 3 mM phenol and/or 1 mM ATP for 15 min at 30 °C in PBS as indicated. A total of 100 μL of 100-fold diluted reactant (100 pM protein with or without 1 mM MgCl$_2$, 3 mM phenol and/or 1 mM ATP) was injected into the biotinylated anti-GFP pAb-coated slide chamber and incubated for 5 min followed by washing with 100 μL of PBS.

To observe the interaction between eGFP-DmpR^WT and Cy5-labelled $\sigma^{54}_{(1-119)}$-CPD, anti-GFP (biotin) goat polyclonal antibody (pAb) (Abcam, #ab6658) was injected into the chambers and incubated for 5 min prior to a wash step with 100 μL of imaging buffer [50 mM HEPES-NaOH pH 7.5, 300 mM NaCl, 5 mM MgCl$_2$, 1% dextrose monohydrate (w/v, Sigma, D9559) and 1 mM Trolox ((±)-6-Hydroxy-2,5,7,8-tetramethylchromane-2-carboxylic acid, Sigma, 238813)]. One microlitre of 10 nM eGFP-DmpR^WT was incubated with 1 mM MgCl$_2$, 3 mM phenol, and/or 1 mM ATPγS for 30 min at 37 °C in the imaging buffer. A total of 100 μL of 100-fold diluted reactant (100 pM protein with 1 mM MgCl$_2$, 3 mM phenol, and 1 mM ATPγS) was injected into the biotinylated anti-GFP pAb-coated slide chamber and incubated for 5 min followed by washing with 100 μL of imaging buffer. A total of 50 μL of 1 nM Cy5 labelled $\sigma^{54}_{(1-119)}$-CPD was incubated in the eGFP-DmpR-coated microfluidic chamber for 5 min followed by washing with 100 μL of imaging buffer supplemented with 0.1 mg/mL glucose oxidase (Sigma, G2133), 4 mg/ml catalase (Roche, 10106810001). A series of EMCCD images were acquired with laboratory-made software with a time resolution of 100 msec. The fluorescence time traces were extracted with an algorithm written using IDL (ITT Visual Information Solutions) that defined the fluorescence spots according to a threshold defined by a Gaussian profile. The extracted time traces were analysed using customized MATLAB programs (MathWorks). The counting of photobleaching steps was performed manually. Stepwise fitting lines in the representative traces were also drawn manually using Illustrator (Adobe).

**Structure determination**. Crystallization was conducted using the sitting-drop vapor diffusion method at 4 °C with DmpR^AD protein (13.5 mg/ml, 1.5 μl) and an equal volume of the crystallization solution (340 mM Na/K-tartrate, 80 mM

glycine, 3 mM AMP-PNP and 10 mM phenol). Before data collection, the crystals were cryocooled to 93 K using a cryoprotectant consisting of mother liquor and 25% glycerol. The diffraction data set was collected using the MX7A synchrotron beamline at the Pohang Accelerator Laboratory (Pohang, Korea). The crystals diffracted to a resolution of 3.4 Å, and the data were collected by 365° rotation of the crystal at 1° intervals. The diffraction data were processed and scaled using HKL2000. The structure was determined by the molecular replacement method using the CCP4 and Phenix suite with the structures of PoxR (PDB ID, 5fru) and NtrC1 (PDB ID, 1ny5) as search models for the sensory and ATPase domains, respectively. The model building and structure refinement were performed using the programs Wincoot and Phenix. Molecular images were produced using Pymol. The Ramachandran statistics for the model are as follows: 94 % of the residues were in the favoured region, 5% of the residues were in the allowed region and 1% of residues were in the outlier region. The crystallographic data that support the findings of this study (PDB ID; 6IY8) are available from the Protein Data Bank. The crystallographic data statistics are summarized in Supplementary Table 2.

**MALS and BN-PAGE**. MALS analysis was performed using a WTC-050S5 SEC column with an in-line Dawn Helios II system and an Optilab T-rEX differential refractometer (Wyatt). DmpR (10 μM), phenol (1 mM) and/or ATP/ATPγS (3 mM) were incubated at 25 °C for 20 min in PBS buffer. After centrifugation, the supernatant was applied to a SEC-MALS system with PBS elution buffer containing 0.5 mM phenol. The data were collected and analysed using ASTRA 6 (Wyatt). Gradient gels (4–16%) were used for BN-PAGE (Novex). To identify factors that might influence the oligomer state of DmpR, 20 μM DmpR, was incubated for 20 min at 25 °C in the presence or absence of 1 mM phenol, 5 mM MgCl$_2$ and/or 3 mM ATP, respectively. To determine change of DmpR tetramer by ATP analogue or UAS containing DNA, 3 mM ATP analogue (AMP-PNP/ATPγS), 10 nM DNaseI (NEB) or 20 μM cognate DNA with the UAS sites were co-incubated with DmpR for 20 minutes prior to BN-PAGE analysis.

**Size exclusion chromatography and dynamic light scattering**. DmpR (20 μM) and phenol (0.5 mM) were incubated at 25 °C for 20 min in PBS buffer. After centrifugation, the supernatant was applied to a SEC or DLS system. SEC analysis was performed using a Superdex 200 increase 10/300 column with an AKTA FPLC system (GE Healthcare). DLS analysis was performed using a Zetasizer Ultra (Malvern), fitted with a 10-mW 632.8 nm laser with scattering angle of 173° in air and set at a 90° scattering angle.

**Molecular docking modelling**. The inactive DmpR dimer was modelled using the A. aeolicus NtrC1 in complex with ADP (PDB ID, 1ny5) as the template. The dimeric NtrC1 structure was truncated so that only the ATPase domain was retained. The docking of ADP to the DmpR^AD structure with loop modelling was performed using the SwissDock server. The initial models were subjected to energy minimization followed by 1 ps of molecular dynamics at 300 K after equilibration. They were finally minimized to a maximum derivative with 1.0 kcal per step using the Discover module in the Insight II program (Accelrys) with the AMBER force field.

**ATPase assay**. The ATPase reactions were initiated by adding 5 mM MgCl$_2$ into a mix containing 200 nM DmpR protein, 50 μM ATP, [γ³²P] ATP (5 Ci/mmol) or/ and 1 mM phenol or/and 1 mM $\sigma^{54}$ in phosphate buffed saline. The reactions (20 μl) were incubated at 25 °C for 20 min and then terminated by the addition of 10 mM EDTA. The radiolabelled reaction products (1.5 μl) were separated with polyethyleneimine-cellulose thin-layer plates (Merck) in 0.325 M phosphate buffer (pH 3.5) and visualised using a FLA-5100 phosphorimager (Fujifilm).

**Isothermal titration calorimetry**. The ITC experiments were conducted using a MicroCal Auto-iTC200 at 25 °C at the Korea Basic Science Institute (KBSI). The DmpR solution (10 μM or 40 μM) in the calorimetric cell was titrated with the phenol ligand (100 μM), cognate DNA with specific UAS sequences (100 μM), or $\sigma^{54}_{(1-119)}$-CPD protein (400 μM) as the injectant. The data were analysed with the MicroCal Origin software package (GE Healthcare).

**Far-Western blot assay**. Purified $\sigma^{54}$ protein (0.25 μg) was resolved by 10% SDS-PAGE and electro-transferred onto PVDF membranes (GE Healthcare). The $\sigma^{54}$ bound to the membrane was refolded by incubation in 6 M~0.1 M guanidine-HCl in AC buffer (10% glycerol, 100 mM NaCl, 20 mM Tris, 0.5 mM EDTA, 1 mM DTT and 0.1% Tween-20) supplemented with 5% milk powder for 3 h at room temperature. Then, the membrane was washed with AC buffer supplemented with 5% milk powder for 2 hours at 4 °C prior to incubation with 500 μg/ml His-tagged DmpR bait protein at 4 °C overnight. The membrane was subsequently washed and incubated for 1 hour with His-tag antibody (Invitrogen, #MA1-21315, 3000-fold dilutions) in phosphate-buffered saline with Tween-20 (PBST) with 3% milk powder. After washing with PBST, the membrane was incubated for 1 hour with the anti-mouse secondary antibody (Sigma, #A3562, 30,000-fold dilutions).

**Reporting summary**. Further information on research design is available in the Nature Research Reporting Summary linked to this article.

## Data availability

The source data underlying Figs. 1f–j, 4b–e and 6b, d, g and Supplementary Figs. 1a–f, 2b, 7b, c, e and 8c are provided as a Source Data file. Coordinates and structure factors have been deposited in the Protein Data Bank (PDB) with the accession code 6IY8. Other data are available from the corresponding authors upon reasonable request.

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

## Acknowledgements

This research was partly supported by the Marine Biotechnology Programme of the Korea Institute of Marine Science and Technology Promotion (KIMST), the Ministry of Oceans and Fisheries (MOF) (No. 20170488), the National Research Fund (NRF-2018R1A2A2A05021648) and the KRIBB Research Initiative. S.-G.L. supported partly by the grant from the National Research Foundation (2018R1A2B3004755). S.K. was partly funded by the European Union's Horizon 2020 research and innovation programme under the Marie Skłodowska-Curie Grant Agreement No. 753528. C.J. was funded by the Foundation for Fundamental Research on Matter (Projectruimte 15PR3188).

## Author contributions

K.-H.P., S.K. and E.-J.W. conceived the study. S.-G.L. and V.S. provided scientific and experimental suggestions. K.-H.P., S.K., S.-J.L., J.-E.C., H.-N.S., A.B.D. and W.-C.A. performed the protein purification and/or crystallization. The structural data analysis and refinement were performed by K.-H.P. and E.-J.W. The biochemical experiments were performed by K.-H.P., S.-J.L., V.V.P. and W.-C.A., and the single-molecule

fluorescence analysis was performed by S.K. and C.J. The manuscript was written by K.-H.P., S.K., V.S. and E.-J.W. with input from all authors.

## Competing interests

The authors declare no competing interests.

## Additional information

**Peer review information** *Nature Communications* thanks the anonymous reviewer(s) for their contribution to the peer reviewa of this work. Peer reviewer reports are available.

