## [Peer Review File · Nature Communications]

Reviewers' comments:

Reviewer #1 (Remarks to the Author):

In this manuscript, Park et al. study the oligomerization of DmpR, a transcriptional enhancer that regulates response to aromatic compounds in *Pseudomonas* sp. The authors use a battery of biochemical, structural and biophysical approaches to demonstrate that DmpR protein forms tetramers. The authors show that the tetrameric assembly is required for ATPase activity and binding to the effector protein, sigma54. The manuscript is well written and the data are convincing. I only have a few minor comments that may strengthen the manuscript.

1. Although the blue native gels provide evidence for protein oligomerization, the single-molecule data presented may benefit from presenting a positive control. What would be the expected distribution for photobleaching steps from a pure tetrameric protein? Can the authors deconvolve the photobleaching data and recapitulate the distribution between dimer and tetrameric species observed in the native page assays? The authors should comment on how the data are classified as precipitates and what fraction of spots are in this category. Is it possible to remove precipitates by say, ultracentrifugation?
2. Does treatment with DNase affect the oligomerization of full-length protein? Does the addition of the cognate DNA sequence affect oligomerization status?
3. Data in Fig. 4b suggests that there is an approx. 400kDa species on blue native gels. The authors should comment on this data in the text. Is a similar species also observed with wt DmpR?
4. Why is the photobleaching data presented in Fig. 6g different from Fig. 1? Specifically, why does immobilization with anti-GFP result in a substantially high fraction of 3-step photobleaching events as compared to immobilization with anti-His antibody?

Ankur Jain

Reviewer #2 (Remarks to the Author):

In this manuscript the authors report the interesting observation that phenol-bound DmpR forms a tetramer and DmpR interacted with the N-terminal peptide of $\sigma 54$ only in the presence of phenol and ATP γ S. This indicates that DmpR-like bEBPs tetramers activate $\sigma 54$ -dependent transcription by using a mechanism distinct from that of hexameric AAA+ ATPases.

Overall, this manuscript is interesting. However there are many important problems remaining to be addressed:

1. The lack of phenol-free DmpR structure makes structural comparison of DmpR with and without phenol binding impossible. If the crystallization is not successful the authors should perform some more experiments to provide more information about the conformational change induced by phenol binding.
2. P4, L20. Is there an equilibrium existing between DmpR dimer and tetramer? Moreover, the supplementary figure 1A and 1B are not consistent. There 1B shows much more tetramer than 1A does even both have phenol. In figure 1A, although more tetramer form in the presence of phenol the effect is not very significant.
3. P5, L4. The presence of a tetrameric subpopulation before the addition of phenol presumably resulted from the binding of aromatic metabolites derived from *E. coli*. Is there any solid evidence that support this statement? Did the authors try to purify phenol-free DmpR to confirm this argument?
4. P5, L6-22. Monomeric DmpR is observed. Does usage of the antibody affects the oligomerization of protein?
5. P6, L6. What is the stoichiometry between DmpR and phenol since this protein shift from dimer

to tetramer during the titration?

6. P7, L11. The binding specificity of phenol should be analyzed. Is phenol the best ligand for DmpR?

7. P8, L12. "ANP-PNP", "AMP-PNP"? How about the KD value?

8. P9, L6. Does the interaction between DmpR and $\sigma 54$ has an effect on the ATPase activity?

9. P11, L9-11. The stoichiometry of the ITC indicates a 1:1 ratio. However Supplementary Figure 7b indicates too weak a band for $\sigma 54$.

10. The authors did not provide data showing how phenol affects DNA binding ability of DmpR. This could be important since transcription depend on DNA binding.

11. The resolution of DmpR is low (3.422\AA). The quality of the structure could be questionable.

**Reviewer #1 (Remarks to the Author):**

**In this manuscript, Park et al. study the oligomerization of DmpR, a transcriptional enhancer**
**that regulates response to aromatic compounds in *Pseudomonas* sp. The authors use a battery of**
**biochemical, structural and biophysical approaches to demonstrate that DmpR protein forms**
**tetramers. The authors show that the tetrameric assembly is required for ATPase activity and**
**binding to the effector protein, sigma54. The manuscript is well written and the data are**
**convincing. I only have a few minor comments that may strengthen the manuscript.**

**Response:** We appreciate the referee's positive comments. Below we provide point-by-point
responses to the questions/points raised.

**1. Although the blue native gels provide evidence for protein oligomerization, the single-**
**molecule data presented may benefit from presenting a positive control. What would be the**
**expected distribution for photobleaching steps from a pure tetrameric protein? Can the authors**
**deconvolve the photobleaching data and recapitulate the distribution between dimer and**
**tetrameric species observed in the native page assays?**

**Response:** We'd like to thank the referee for suggesting this important control. To find out the
distribution of photobleaching steps from pure dimeric and tetrameric fluorescent proteins, we
constructed an eGFP-fused Cas1 (eGFP/Cas1) dimer and an eGFP/Cas1 tetramer complexed with a
Cas2 dimer. When Cas1 protein is solely expressed in *E. coli* BL21-AI cells, which lack all *cas* genes,
it is naturally assembled as a dimer^{1, 2} (Revision Fig. 1a). When both Cas1 and Cas2 proteins are
expressed, two dimers of Cas1 and one dimer of Cas2 form a heterohexamer³. We have confirmed
heterohexameric eGFP/Cas1-Cas2 complexes that contain four eGFP motifs (Revision Fig. 1b) and
have pooled eGFP dimer and tetramer fractions for single-molecule photobleaching (SMPB) assays.
Of note, we measured 85% eGFP maturation efficiency for both pooled eGFP/Cas1 and eGFP/Cas1-
Cas2 complexes using spectrophotometry.

Revision Figure 1. Size-exclusion chromatography and SDS-PAGE of Cas1, Cas1-Cas2 and eGFP/Cas1-Cas2 complexes

a. Size-exclusion fast performance liquid chromatography (SEC) of StrepTactin purified N-terminally StrepII-tagged Cas1 (upper) and Cas1-Cas2 (lower). Bracketed fractions were pooled and resolved in SDS-PAGE (NuPAGE). Cas1 (~36 kDa) and Cas2 (~11 kDa) are indicated. **b.** SEC of StrepTactin purified N-terminally StrepII and eGFP tagged Cas1 and Cas2 complex. Bracketed fractions were resolved in SDS-PAGE (NuPAGE). eGFP/Cas1 (~60 kDa) and Cas2 (~11 kDa) are indicated. Pooled fractions, which contain heterohexameric (tetrameric eGFP) eGFP/Cas1-Cas2 and dimeric (dimeric eGFP) eGFP/Cas1 complexes, are indicated as a dotted orange regions for SMPB assays.

For the SMPB assay with eGFP dimer (eGFP-Cas1 homodimer) and tetramer (eGFP/Cas1-Cas2 heterohexamer), we surface-immobilized the complexes using biotinylated anti-GFP polyclonal antibody through biotin-streptavidin conjugation on a biotin-PEGylated quartz slide surface in a microfluidic chamber. Stepwise bleaching signals from eGFP were recorded using total internal

reflection fluorescence (TIRF) microscopy (Revision Fig. 2a). One to four steps were observed from
 individual eGFP fluorescence time traces (Revision Fig. 2b).

- • Dimeric eGFP signals showed 20% of one step and 49% of two steps (Revision Fig. 2c). The
 relative fraction of the two-step photobleaching population is 0.71, which agrees with the
 statistical expectancy of such a population from the 85% eGFP maturation ($0.85 \times 0.85 = 0.72$).
- • Tetrameric eGFP exhibited 4%, 7%, 22% and 35% for one, two, three and four steps,
 respectively (Revision Fig. 2c). The relative fraction of each is 0.06, 0.10, 0.32 and 0.51,
 respectively, which also agree with the statistical expectancy from the 85% eGFP maturation
 (one step, 0.01; two steps, 0.10; three steps, 0.37; four steps, 0.52).

**Revision Figure 2. Single-molecule photobleaching assays of eGFP/Cas1 and eGFP/Cas1-Cas2**
 **complexes**

**a.** A representative EMCCD image of eGFP/Cas1 (left) and eGFP/Cas1-Cas2 Monomeric, dimeric,
trimeric and tetrameric signals are indicated. Asterisks represent the signal from presumable protein
aggregates. **b.** Representative time trajectories of the eGFP emission signals. The stoichiometry of
the eGFP oligomers was determined by counting the number of eGFP photobleaching steps. Light
blue lines are eGFP emission traces. Pink lines represent stepwise fits of the traces. **c.** Distribution of
photobleaching steps of eGFP signals. Events with more than 8 photobleaching steps were
categorized as aggregates (A).

**Can the authors deconvolve the photobleaching data and recapitulate the distribution between**
**dimer and tetrameric species observed in the native page assays?**

**Response:** We thank the referee for suggesting this analysis. To deconvolute the distribution pattern of
our photobleaching data, we estimated the expected distribution of a ratio between apparent dimeric
and tetrameric eGFP populations ($r = \text{dimer/tetramer}$) while considering that eGFP maturation rate is
0.85 (Revision Fig. 3a). The photobleaching steps of purely dimeric and tetrameric eGFP populations,
which were positive controls in Revision Fig. 2, were overlaid on the simulated distributions. It shows
that they correspond to the distributions of $r = 0.0$ and 1.0 , respectively, meeting our expectation on
the positive controls (Revision Fig. 3a).

Likewise we recapulated the ratio (r) for Fig. 1f-j (Revision Fig. 3b). The distributions for
no-phenol regardless of the presence of ATP or Mg^{2+} correspond to the distribution of $r = 0.4$; the
distribution for phenol, $r = 0.8$. This agrees with the ratio between dimers and tetramers in the native
gel electrophoresis shown in Supplementary Fig. 1a. In addition, when we fitted the distributions from
Fig. 6d and 6g, we saw that it corresponded to $r = 1.0$ as expected (Revision Fig. 3c and 3d). Overall,
the deconvolution using the positive control data could recapitulate the ratio between dimer and
tetramer of eGFP, further supporting our model.

**Revision Figure 3. Single-molecule photobleaching assays of eGFP/Cas1 and eGFP/Cas1-Cas2**
 **complexes**

**a.** Distributions of the photobleaching steps of dimeric eGFP (eGFP/Cas1₂, red) and tetrameric eGFP
 (eGFP/Cas1₄-Cas2₂, blue), related to Revision Fig. 2c. Ten different ratio (r) between dimer and
 tetramers were simulated. **b.** Distributions of eGFP/DmpR photobleaching signals with or without
 phenol, ATP and MgCl₂ (purple), related to Fig. 1f. **c.** Distributions of eGFP/DmpR photobleaching
 signals (purple), related to Fig. 6d. **d.** Distributions of eGFP/DmpR photobleaching signals (purple),
 related to Fig. 6g. (**a-d**) Each distribution is overlaid on the distributions of the photobleaching steps
 of each theoretical dimer/tetramer ratio at 0.1 intervals with 85% of a fluorophore maturation rate.

**The authors should comment on how the data are classified as precipitates and what fraction of**
 **spots are in this category. Is it possible to remove precipitates by say, ultracentrifugation?**

**Response:** Precipitates are molecules that show events with more than 8 photobleaching steps. The
 signal intensities are abnormally high or hardly exhibit distinct photobleaching steps (Fig. 1c and 1e).

As suggested by the referee, we repeated the SMPB assay with eGFP-tagged DmpR^{WT} after
 removing clotted precipitates by ultracentrifugation followed by size-exclusion fast performance
 liquid chromatography. Nevertheless, we have obtained almost an unaltered result that showed 15–25%

fractions of aggregates (Fig. 1f–j). We speculate that this might be attributed to antibody aggregates
that could be triggered by partial unfolding of their domains, leading to monomer-monomer
association followed by nucleation and growth during purification, concentration and more critically
biotin labelling procedures⁴.

**2. Does treatment with DNase affect the oligomerization of full-length protein? Does the**
**addition of the cognate DNA sequence affect oligomerization status?**

**Response:** In view of this comment, we performed additional experiments to determine whether
DNase treatment or binding to DNA containing the target sites for DmpR (UASs, Upstream
Activating Sites) influences the oligomeric state of DmpR. Within these experiments, reaction sample
of DmpR (20 μM) with either DNaseI (NEB, 10 nM) or a 54 bp DNA fragment bearing the two UASs
for DmpR binding in their native configuration (20 μM) were incubated for 20 minutes in presence of
phenol prior to analysis by BN-PAGE (Supplementary Fig. 1d, right). DNAase treatment did not
affected phenol induced oligomerization of DmpR. When we analyzed the binding of the DNA to
DmpR, it exhibits a high binding affinity ($K_d \sim 387\text{nM}$) as measured by ITC (Supplementary Fig. 1e).
Nevertheless, the phenol induced tetramer formation of DmpR were not affected by the cognate DNA.
We included these results in the revised manuscript (Page 5 / Line 1) with the data presented in
Supplementary Fig. 1d & 1e.

**3. Data in Fig. 4b suggests that there is an approx. 400kDa species on blue native gels. The**
**authors should comment on this data in the text. Is a similar species also observed with wt**
**DmpR?**

**Response:** The band corresponding to ~ 400 kDa shown in Fig. 4b is thought to be an artifact of the
DmpR^{ΔS} tetramer, a potential octamer (~ 350 kDa) consisting of two tetramers. We could not detect
any band of the higher molecular size in the BN-PAGE of wild type DmpR. Deletion of the sensory
domain as in DmpR^{ΔS} would likely expose the coiled coil region of the B-linkers and thus uncover
some hydrophobic region on the surface of ATPase tetrameric domains. Coiled coil architecture of the
alpha helices, protruding from the ATPase domains, probably interact with another coiled coil B-
linkers forming a non-native oligomer artefact of the DmpR^{ΔS} truncate. Similar non-native oligomers
have previously been reported for truncates of other NtrC family members that likewise lack their
sensory domains^{5,6}. Hence, accordingly, we included a short explanatory description in the revised
manuscript (Page 9 / Line 9 and Fig. 4b).

**4. Why is the photobleaching data presented in Fig. 6g different from Fig. 1? Specifically, why**
**does immobilization with anti-GFP result in a substantially high fraction of 3-step**
**photobleaching events as compared to immobilization with anti-His antibody?**

**Response:** We thank the referee for pointing this out. In our first submission, we did the SMPB assay
with eYFP-tagged DmpR^{WT} using anti-His system for Fig. 1 and with eGFP-tagged DmpR^{WT} using
anti-GFP for Fig. 6. We found that the use of anti-GFP resulted in clearer fluorescence signals, less
background fluorescence, and fewer aggregates. For consistency and better statistics, we have
repeated the experiments for Fig. 1 using eGFP-tagged DmpR^{WT} and anti-GFP. The new results show
a higher fraction of 3-step events. We have updated Fig. 1 with this new data in the revised manuscript.

**Reviewer #2 (Remarks to the Author):**

**In this manuscript the authors report the interesting observation that phenol-bound DmpR**
**forms a tetramer and DmpR interacted with the N-terminal peptide of σ 54 only in the presence**
**of phenol and ATP γ S. This indicates that DmpR-like bEBPs tetramers activate σ 54-dependent**
**transcription by using a mechanism distinct from that of hexameric AAA+ ATPases.**

**Overall, this manuscript is interesting. However there are many important problems remaining**
**to be addressed:**

**Response:** We appreciate the helpful comments and suggestions of the referee and have made changes
accordingly.

**1. the lack of phenol-free DmpR structure makes structural comparison of DmpR with and**
**without phenol binding impossible. If the crystallization is not successful the authors should**
**perform some more experiments to provide more information about the conformational change**
**induced by phenol binding.**

**Response:** According to the suggestion, we performed additional experiments that are included in the
revised manuscript. First, we analyzed the conformational change by dynamic light scattering and also
performed size exclusion chromatography in presence or absence of phenol. As clearly seen in the
Supplementary Fig. 1b & 1c, DmpR exhibits a significant shift of its molecular weight from the
approximate dimer size in the absence of phenol to that of the tetramer in presence of phenol, both in
the dynamic light scattering and in the size exclusion chromatography. We included these data in the
revised manuscript (Page 4 / Line 23).

It is a widely accepted view with many supporting evidences that DmpR exhibits
conformational changes induced by phenol binding in order to initiate transcription⁷. DmpR shares
high sequence homology with other aromatic responsive bEBPs, such as XylR, TouR, PoxR and
MopR, and these subgroups are known to transition from inactive dimers to active oligomers upon
binding of the aromatic effectors⁸. A recent report showed that the central AAA⁺ domain of DmpR
forms a homodimer with a face-to-face orientation⁹, clearly supporting the conformational change of
DmpR. Here, in this study, we demonstrated that DmpR changes its oligomeric state from dimer to
tetramer by introducing the single molecular photobleaching technique, in addition to determining the
x-ray structure. We have tried extensively to obtain the DmpR crystal in the absence of phenol, but
unfortunately all trial have failed. We believe that lack of crystal formation in absence of the phenol
actually indicates the existence of a different DmpR conformation from the ligand bound form.

**2. P4, L20. Is there an equilibrium existing between DmpR dimer and tetramer? Moreover, the**
**supplementary figure 1A and 1B are not consistent. There 1B shows much more tetramer than**
**1A does even both have phenol. In Figure 1A, although more tetramer form in the presence of**
**phenol the effect is not very significant.**

**Response:** We do not have any evidence for the existence of equilibrium between the dimer and the
tetramer of DmpR. According to the ITC binding measurement, phenol binds to DmpR with a
relatively high affinity (Kd 12~16 μ M). We observed a gradual increase of the tetramer form over
time after phenol addition, which indicates either irreversible phenol binding or stable formation of
the tetramer. The discrepancy in the bands between Supplementary Fig. 1a and 1d (before revision, 1b)
resulted from different experimental condition. In Supplementary Fig. 1a, where the purpose of the
experiment is to analyze the effect of individual factor on the oligomerization of DmpR, each
component was added and incubated for 20 min before being applied to Blue-Native PAGE. Once
ATP was determined to have no effect on tetramer formation, we analyzed the comparative effect of
each ATP analogue on the oligomer formation since those analogues are not cleavable and were used
further in crystallization and functional study. In Supplementary Fig. 1d, the DmpR protein was first
incubated with phenol for 20 min and then the ATP molecule or its analogue was added in the reaction
sample for the assay. The higher population of tetramer in the Supplementary Fig. 1d is consistent to
the observation of a gradual increase of the tetramer pool over time. To avoid confusion and clarify
this issue, we changed the label of Supplementary Fig. 1d in the revised manuscript with added details
on the methodology used (Page 19 / Line 4).

**3. P5, L4. The presence of a tetrameric subpopulation before the addition of phenol presumably**
**resulted from the binding of aromatic metabolites derived from E. coli. Is there any solid**
**evidence that support this statement? Did the authors try to purify phenol-free DmpR to**
**confirm this argument?**

**Response:** The host strain of *E.coli* BL21, used for expression of DmpR protein in this study, is
known to synthesize various aromatic metabolites such as 4-aminobenzolate, benzoyl-CoA,
phenylacetic acid and phenylpropionic acid¹⁰. Despite extensive efforts, we were not able to purify a
complete pure form of apo DmpR from the *E. coli* expression system. There are many previous
reports of aromatic compound incorporation during purification of aromatic hydrocarbon binding
proteins from the *E. coli* system, such as toluene binding protein (TodS), phenol binding proteins
(MopR and PoxR) and halogenated benzene binding in T4 lysozyme^{11, 12, 13}. We now include those
references in the revised manuscript (Page 5 / Line 7).

**4. P5, L6-22. Monomeric DmpR is observed. Does usage of the antibody affects the**
**oligomerization of protein?**

**Response:** We speculate that the observation of the monomeric DmpR in the single molecule data is
attributed to truncation and/or partial maturation of the fluorescent proteins eYFP/eGFP. (Please check
our answer to Question 1 by Reviewer 1). In addition, the SDS-PAGE of the recombinant eYFP-
DmpR sample (used in the initial submission) showed the presence of eYFP proteins without DmpR
conjugation (Revision Fig. 4a). This probably resulted from autocleavage or abortive translation of the
proteins during overexpression and purification in *E.coli* system¹⁴. For the preparation of eGFP-
DmpR protein, we purified the eGFP-tagged protein by affinity chromatography followed by size-
exclusion chromatography (Revision Fig. 4b, upper). (Note that we switched eYFP with eGFP for the
better photophysical property of eGFP). We further used fractions that contain full length eGFP-
DmpR and discarded the cleaved products that contains eGFP protein fragments (Revision Fig. 4b,
lower). This purification resulted in a lower percentage of one-step photobleaching population.

Given the 85% maturation rate of eGFP, the remaining one-step bleaching likely resulted
from partial maturation of the fluorescent protein. In the Revision Fig. 2, the dimeric eGFP signals
showed 20% of one step and 49% of two steps. The relative fraction of the two-step photobleaching
population is 0.71, which agrees with the statistical expectancy of such a population from the 85%
eGFP maturation ($0.85 \times 0.85 = 0.72$). Likewise, tetrameric eGFP exhibited 4%, 7%, 22% and 35%
for one, two, three and four steps, respectively (Revision Fig. 2c). The relative fraction of each is 0.06,
0.10, 0.32 and 0.51, respectively, which agree with the expectancy from the 85% eGFP maturation
(one step, 0.01; two steps, 0.10; three steps, 0.37; four steps, 0.52). In the revised manuscript, we have
modified the text as follows. “*One-step bleaching from dimeric eGFP-DmpR^{WT} and less-than-four-*
*step bleachings from tetrameric eGFP- DmpR^{WT} could originate from incomplete eGFP maturation.*
*The eGFP maturation was estimated to be 85% from the ratio between a protein concentration*
*measured from the 280-nm absorbance and an eGFP fluorophore concentration measured from 488-*
*nm absorbance”.*

**Revision Figure 4. Quality check for eYFP-DmpR and eGFP-DmpR**
**protein.**

**a.** SDS-PAGE (NuPAGE, Invitrogen™) of eYFP-DmpR used in the single-
molecule photobleaching (SMPB) assays (first manuscript). Full length of
eYFP-DmpR protein (94.5 kDa) and both truncated eYFP (35 kDa) and
DmpR (55 kDa) are visible.

**b.** Size-exclusion chromatography (SEC) of Ni²⁺-NTA purified eGFP-DmpR
proteins (upper). Fractions #1 and #2 were resolved in SDS-PAGE
(NuPAGE), which shows full length of eGFP-DmpR protein (94.5 kDa) and
DmpR (55–70 kDa) fragments in Fraction #1 and truncated eGFP (28–35
12 kDa) in Fraction #2. The fractions used for SMPB assays (revised
manuscript) are indicated as a dotted orange region.

**5. P6, L6. What is the stoichiometry between DmpR and phenol since this protein shift from**
**dimer to tetramer during the titration?**

**Response:** To address this question, we measured the binding stoichiometry between DmpR^{WT} and
phenol by ITC analysis. The titration curve with a tetramer molecular weight of 264 kDa ($n = 0.258 \pm$
0.007) indicates the stoichiometry between DmpR tetramer and phenol as 1:4 ratio, that corresponds
to 1:1 ratio of DmpR monomer and phenol molecule. We included the stoichiometry data in
Supplementary Fig. 2a of the revised manuscript.

**6. P7, L11. The binding specificity of phenol should be analyzed. Is phenol the best ligand for**
**DmpR?**

**Response:** The substrate binding specificity of DmpR has been previously reported¹⁵ and the
luciferase coupled transcriptional activity of DmpR has also been analyzed and well documented
against 34 aromatic compounds⁷. The transcriptional promoting activity of DmpR was found to be
highest when phenol and 2-methylphenol were used as substrates and DmpR has a high binding
affinity for phenol ($K_d \sim 16 \mu\text{M}$)¹⁶. These data indicates that the best ligand for DmpR is phenol. We
measured the binding affinity of DmpR^{ΔD} protein toward phenol ($K_d \sim 12 \mu\text{M}$) by ITC analysis and
included it in the revised manuscript (Supplementary Fig. 2a) together with references for the binding
specificity of DmpR (Page 3 / Line 9).

**7. P8, L12. “ANP-PNP”, “AMP-PNP”? How about the KD value ?**

**Response:** ‘ANP-PNP’ is a typo that we have corrected in the revised manuscript (Page 8 / Line 20).
The K_d value of AMP-PNP toward the ATPase domain of NtrC was previously reported to be
approximately $12 \mu\text{M}$ ^{17, 18}. Since the ATPase domain of DmpR shows high structural similarity
(r.s.m.d. $\sim 1.2 \text{ \AA}$) to the ATPase domain of NtrC that has a sequence homology over 75%, DmpR is
expected to exhibit the similar binding affinity toward the AMP-PNP molecule.

**8. P9, L6. Does the interaction between DmpR and σ^{54} has an effect on the ATPase activity?**

**Response:** To answer this question, we conducted an experiment to determine whether the interaction
of DmpR^{WT} and the sigma factor σ^{54} affects the ATPase activity. The ATPase assay was performed
under the same conditions as the previous assay using TLC analysis. When the DmpR^{WT} was
measured for the ATPase activity in presence of the sigma factor σ^{54} , DmpR exhibited a similar level
of activity with only marginal increase of 3% as compared to the DmpR^{WT} alone, indicating the

interaction between DmpR and σ^{54} does not affect the ATPase activity (Supplementary Fig. 7e). We
believe that DmpR utilizes ATP hydrolysis to interact with the σ^{54} transcription factor. We included
this result in the revised manuscript (Page 11 / Line 14).

**9. P11, L9-11. The stoichiometry of the ITC indicates a 1:1 ratio. However Supplementary**
**Figure 7b indicates too weak a band for σ^{54} .**

**Response:** Far-western blot assays examine protein interactions based on a limited amount of
correctly refolded protein after separation by SDS-PAGE and transfer to a filter. Many interaction data
by Far-western blot does not correlate with the stoichiometry and show relatively weak bands¹⁹ since
this method provides only qualitative information about the protein-protein interaction. We understand
that the positive control used in the Figure may have confused the data analysis, therefore we
modified the label of the positive control in the revised manuscript (Supplementary Fig. 7b).

**10. The authors did not provide data showing how phenol affects DNA binding ability of DmpR.**
**This could be important since transcription depend on DNA binding.**

**Response:** We agree and thus further investigated how phenol affects the DNA binding ability of
DmpR using ITC analysis (as referred to in our response to reviewer 1, point 2). Since the DmpR-
regulated Po promoter of *P. putida* has two palindromic UAS elements separated by 20 base pairs, we
designed a cognate 54 bp linear DNA containing these two UAS elements. When we examined the
interaction of DmpR^{WT} to the DNA fragment in presence of phenol, the result showed a standard
binding curve with a Kd ~387 nM. When we examined the binding to DNA in absence of phenol, it
showed a binding affinity of Kd ~476 nM – only a marginal change as compared to the binding in
presence of phenol (Supplementary Fig. 1e). This result suggests that the DNA binding ability of
DmpR is not significantly affected by phenol. Since the DNA binding domains are located away from
the phenol sensory domains in the structure, and the DNA binding domains could bind to DNA by
themselves, the binding of phenol to the sensory domain of DmpR is unlikely to affect the binding
affinity of the DNA binding domain. We speculate that the tetramerization of the DmpR may result in
a topology change of the DNA to expose the nucleotide strand for accessibility, while interaction with
σ^{54} of the transcription machinery initiates transcriptional activation. We included these results in the
revised manuscript (Page 5 / Line 2) with a Supplementary Fig. 1e.

**11. The resolution of DmpR is low (3.422Å). The quality of the structure could be questionable.**

**Response:** Although the resolution of DmpR may not be high enough for detailed atomic interactions,
we believe the 3.422 Å resolution structure provides essential information regarding the
conformational changes and the oligomer formation, which is sufficient to elucidate the important
functional mechanism in question. Previously we determined the sensory domain of PoxR, a structural
homologue of DmpR, complexed with phenol ligand in high resolution (1.85 Å)¹² and the detailed
atomic interaction of phenol inside the PoxR domain is expected to be similar to that of DmpR, due to
its high sequence homology (44%).

**References**

- 1. Babu M, *et al.* A dual function of the CRISPR-Cas system in bacterial antiviral immunity and
DNA repair. *Mol Microbiol* **79**, 484-502 (2011).
2. Wiedenheft B, Zhou K, Jinek M, Coyle SM, Ma W, Doudna JA. Structural basis for DNase
activity of a conserved protein implicated in CRISPR-mediated genome defense. *Structure* **17**,
904-912 (2009).
3. Nunez JK, Kranzusch PJ, Noeske J, Wright AV, Davies CW, Doudna JA. Cas1-Cas2 complex
formation mediates spacer acquisition during CRISPR-Cas adaptive immunity. *Nat Struct Mol*
*Biol* **21**, 528-534 (2014).
4. Li W, Prabakaran P, Chen W, Zhu Z, Feng Y, Dimitrov DS. Antibody Aggregation: Insights
from Sequence and Structure. *Antibodies (Basel)* **5**, (2016).
5. Batchelor JD, Sterling HJ, Hong E, Williams ER, Wemmer DE. Receiver domains control the
active-state stoichiometry of Aquifex aeolicus sigma54 activator NtrC4, as revealed by
electrospray ionization mass spectrometry. *J Mol Biol* **393**, 634-643 (2009).
6. Rippe K, Mucke N, Schulz A. Association states of the transcription activator protein NtrC
from *E. coli* determined by analytical ultracentrifugation. *J Mol Biol* **278**, 915-933 (1998).
7. Shingler V, Moore T. Sensing of aromatic compounds by the DmpR transcriptional activator
of phenol-catabolizing *Pseudomonas* sp. strain CF600. *J Bacteriol* **176**, 1555-1560 (1994).
8. Bush M, Dixon R. The role of bacterial enhancer binding proteins as specialized activators of
sigma54-dependent transcription. *Microbiol Mol Biol Rev* **76**, 497-529 (2012).
9. Seibt H, Sauer UH, Shingler V. The Y233 gatekeeper of DmpR modulates effector-responsive
transcriptional control of sigma(54) -RNA polymerase. *Environ Microbiol*, (2019).

- 10. Patnaik R, Liao JC. Engineering of Escherichia coli central metabolism for aromatic
metabolite production with near theoretical yield. *Appl Environ Microbiol* **60**, 3903-3908
(1994).
- 11. Koh S, *et al.* Molecular Insights into Toluene Sensing in the TodS/TodT Signal Transduction
System. *J Biol Chem* **291**, 8575-8590 (2016).
- 12. Patil VV, Park KH, Lee SG, Woo E. Structural Analysis of the Phenol-Responsive Sensory
Domain of the Transcription Activator PoxR. *Structure* **24**, 624-630 (2016).
- 13. Liu L, Baase WA, Matthews BW. Halogenated benzenes bound within a non-polar cavity in
T4 lysozyme provide examples of I...S and I...Se halogen-bonding. *J Mol Biol* **385**, 595-605
(2009).
- 14. Wei J, Gibbs JS, Hickman HD, Cush SS, Bennink JR, Yewdell JW. Ubiquitous
Autofragmentation of Fluorescent Proteins Creates Abundant Defective Ribosomal Products
(DRiPs) for Immunosurveillance. *J Biol Chem* **290**, 16431-16439 (2015).
- 15. O'Neill E, Sze CC, Shingler V. Novel effector control through modulation of a preexisting
binding site of the aromatic-responsive sigma(54)-dependent regulator DmpR. *J Biol Chem*
**274**, 32425-32432 (1999).
- 16. O'Neill E, Ng LC, Sze CC, Shingler V. Aromatic ligand binding and intramolecular signalling
of the phenol-responsive sigma54-dependent regulator DmpR. *Mol Microbiol* **28**, 131-141
(1998).
- 17. Dey S, Biswas M, Sen U, Dasgupta J. Unique ATPase site architecture triggers cis-mediated
synchronized ATP binding in heptameric AAA+-ATPase domain of flagellar regulatory
protein FlrC. *J Biol Chem* **290**, 8734-8747 (2015).
- 18. Chen B, *et al.* ATP ground- and transition states of bacterial enhancer binding AAA+ ATPases
support complex formation with their target protein, sigma54. *Structure* **15**, 429-440 (2007).
- 19. Wu Y, Li Q, Chen XZ. Detecting protein-protein interactions by Far western blotting. *Nat*
*Protoc* **2**, 3278-3284 (2007).

Reviewers' comments:

Reviewer #2 (Remarks to the Author):

The authors have addressed many technical issues in the new version of manuscript. This makes things better. However there are still several significant concerns to be clarified.

1. ATP or its analogues are not included in the structure. Since ATP is critical for the transcription the lack of ATP make it difficult to draw the conclusion that the tetramer is an active form. Did the authors try negative-stain electron microscopy to see if ATP trigger any conformational change or formation of hexamer?

2. Why does DmpR form tetramer while other AAA+ ATPase transcriptional regulators are hexamer? There should be some structural basis for the difference.

3. The author did not discuss the significant inconsistency between this work and the result reported by ref 13. In ref 13 Wikstrom et al mentioned that both tetramer and hexamer had been observed and only the hexamer is active.

4. The PDB access number for this structure should be 6IY8 rather than 6IYB.

Reviewer #2 (Remarks to the Author):

The authors have addressed many technical issues in the new version of manuscript. This makes
things better. However there are still several significant concerns to be clarified.

**Response:** We appreciate the referee's positive comments. Below we provide point-by-point
responses to the questions/points raised.

1. ATP or its analogues are not included in the structure. Since ATP is critical for the transcription the
lack of ATP make it difficult to draw the conclusion that the tetramer is an active form. Did the
authors try negative-stain electron microscopy to see if ATP trigger any conformational change or
formation of hexamer?

**Response:** In order to address this point and to visualize the overall shape of the DmpR-phenol
complex in presence of ATP, we incubated DmpR^{WT} with phenol, ATP and Mg²⁺ and examined the
negatively stained samples using transmission electron microscopy (TEM). The averaged 2-
dimensional (2D) class images of DmpR^{WT} showed a significantly heterogeneous morphology
(**Revision Figure 1A**). Nevertheless, the most prominent class averages exhibited molecules of ~165
Å × ~120 Å with a twisted rod-like shape and “stubby arms”. Some minor class averages with a
smaller size (~90 Å × ~110 Å) were also observed. Given the size of the tetrameric DmpR^{ΔD} crystal
structure [150 Å (horizontal) × 75 Å (vertical)] (**Revision Figure 1B**), the kinked or twisted rod-
shaped molecules of the largest population likely correspond to the tetramer DmpR^{WT}-phenol complex
with the stubby arms being the DNA binding domain (**Revision Figure 1C**). This morphological form
is clearly different from the conventional ring-shaped hexamer structure of AAA⁺ ATPases¹. The
various kinked and twisted shapes of the complexes presumably resulted from the asymmetric
geometry of the protein bound to phenol (Figure 5, Supplementary Figure 4, as described in the
discussion section, page 13 /line 13) with flexible DNA binding domain pairs attached at both side.
Since SMPB analysis showed a distribution of tetramers and dimers under the same conditions, we
assume that the smaller particles (~90 Å × ~110 Å) are likely inactive DmpR dimers.

Revision Figure 1 Electron microscopy analysis of DmpR

(A) 2D image classification of DmpR of various shapes.

(B) Crystal structure of DmpR^{ΔD}

(C) The upper panels show a selection of 2D class averages from three different views. Arrows indicate the putative position of the DNA binding domains, which are highlighted in dark gray in the DmpR models below. The box sizes in the 2D class averages are 180×180 pixels (mask diameter, 256 \AA).

Generally, a structure observed in a crystal represents a conformation trapped in the lattice that
allows stable packing during crystallization. Previously we pointed out the asymmetric phenol
occupancy in the crystal structure of the sensory domain with a weak phenol density in one of the
subunits in the dimer (Figure 5, Supplementary Figure 4). The signal relaying mechanism of DmpR
potentially involves dynamic changes of the angle and the orientation of the ATPase domains in the
tetramer architecture (as described in the discussion section, page 14 /line 8). Given the observation of
the weak phenol density and the asymmetric monomer conformation, it is plausible that the altered
conformation in one of the monomers in the dimer may have resulted in a weak interaction with ATP
molecule during the crystallization process; hence, yielding the absence of ATP molecule in the
crystal. We speculate that complete binding of phenol molecules with full occupancy of all four
subunits in the tetramer may not be favourable for crystallization, but that this dynamic conformation
could induce sufficient binding affinity for ATP in the tetramer structure. Nevertheless, the
observation that the sigma54 peptide interacted predominantly with the tetrameric form of DmpR only
in the presence of ATP analogues (Figure 6, Supplementary Figure 8) clearly demonstrates that the
tetrameric architecture is an active oligomer form and a necessary step for the bEBP to initiate
transcription.

Because the data in Revision Figure 2 does not alter the conclusions of this already data-heavy
manuscript, we have limited changes to the manuscript to text highlighting the dynamic tetramer
conformation of DmpR. The text now included an additional description as follows: "Given the
asymmetric configuration between two monomers in a dimer and the absence of ATP molecule in the
crystal structure, the dynamic DmpR tetramer probably undergo conformational change in the process
of binding and/or hydrolysing ATP that accompanies its interaction with σ^{54} . (Page 13/Line 7-10)

2. Why does DmpR form tetramer while other AAA+ ATPase transcriptional regulators are hexamer?

There should be some structural basis for the difference.

**Response:** As detailed below we believe that the structure of the sensory domain may provide the key
to this difference. Control of typical ring-like hexameric bEBP such as NtrC and other bEBPs of two-
component system is via phosphorylation of their sensory domains. In contrast, control of single-
component DmpR and its homologues is via binding of a ligand (e.g. phenol) and they exhibit a
tetrameric arrangement as described in this study. The ATPase domains of AAA⁺ proteins are highly
conserved and well known as important determinants of stacking for oligomer interaction. In the
DmpR structure, the ATPase domains also contribute significantly to tetramer formation. However,
we noticed that the key determinant to the tetrameric arrangement resides in the unique architecture of
the sensory domains. The two sensory domains in a DmpR dimer exhibits a tight interaction between
monomers with an interlocked architecture in which the N-terminal extension intertwine to make a
complementary interaction with another monomer, forming an antiparallel beta-sheet and 2-fold
symmetry at the center (Figure 2). This tight interlocked configuration is also observed in the sensory
domain structures of DmpR-like PoxR and MopR^{2,3}. This contrasts the sensory domains of hexameric
NtrC family members that have a structure in which each subunit can exist independently as a
monomer or in a monomer-dimer equilibrium in solution^{4,5}. The monomeric architecture of the
phosphorylation-responsive sensory domains of NtrC-like bEBPs could freely move and thus
participate in formation of a ring-like oligomer, whereas the tight dimeric geometry of the DmpR
sensory domain imposes a conformational constraint, causing a steric hindrance for a ring-like
hexameric arrangement. We propose that the unique intertwined structure of the sensory domain is the
determinant factor setting DmpR and its homologues apart from the other typical hexameric AAA⁺
ATPases. Hence, in the light of this comment, we now highlight the structural basis of the DmpR
tetramer formation in the discussion section as follows. “The tightly interlocked sensory domains,
which are also observed in the PoxR and MopR structures^{11,12}, may also be the key structural element
that would prevent hexamer formation and thus set DmpR and its homologues apart from the other
typical hexameric AAA⁺ ATPases (page 14/ line 4-7).

3. The author did not discuss the significant inconsistency between this work and the result reported
by ref 13. In ref 13 Wikstrom et al mentioned that both tetramer and hexamer had been observed and
only the hexamer is active.

**Response:** The apparent tetramer and hexamer subunit configurations of DmpR described in the ref
13 paper (Wikstrom et al) were based on calculations from ultra-centrifugation sedimentation through
glycerol gradients using globular proteins as standards. Protein mass estimates from such analysis,
which are dependent on both the partial specific volume (stokes radius / shape) and density of the
protein, only correlates well if the subject molecule also has a globular shape⁶. However, as observed
in the negative staining electron microscopy as well as in the crystal structure, the shape of DmpR
tetramer is far from globular. Hence, the kinked or twisted rod-shaped conformations of the DmpR
dimers and tetramers are likely to lead to deviation from the molecular mass as determined from
globular protein standards and ultra-centrifugation. In contrast, the multi-angle light scattering
(MALS) technique used in this study (Figure 4C, Supplementary Figure 1F) measures the light
scattered by a sample into a plurality of angles, determining the absolute molar mass. MALS is widely
considered as one of the most accurate methods to determine the molecular weight of protein. We
speculate that the peaks thought to be tetramer and hexamer in the ref 13 actually correspond to a
dimer and tetramer of DmpR, respectively. In that case, the active “hexamer” form is in fact an active
tetrameric DmpR complex, consistent to our study. Based on the MALS and the SMPB data as well as
the crystal structure, there is no doubt that DmpR forms a tetramer in the presence of phenol and ATP.
Because the authors of ref13 pointed out the assumption of a globular shape, at this juncture we have
not modified the revised text to include any discussion of this point. However, we are willing to if
requested to do so.

4. The PDB access number for this structure should be 6IY8 rather than 6IYB.

**Response:** We have corrected it to 6IY8 in the revised manuscript (Page 19 / Line 1).

**References**

- 1. Rappas M, *et al.* Structural insights into the activity of enhancer-binding proteins. *Science* **307**,
1972-1975 (2005).
- 2. Patil VV, Park KH, Lee SG, Woo E. Structural Analysis of the Phenol-Responsive Sensory
Domain of the Transcription Activator PoxR. *Structure* **24**, 624-630 (2016).
- 3. Ray S, Gunzburg MJ, Wilce M, Panjekar S, Anand R. Structural Basis of Selective Aromatic
Pollutant Sensing by the Effector Binding Domain of MopR, an NtrC Family Transcriptional
Regulator. *ACS Chem Biol* **11**, 2357-2365 (2016).
- 4. De Carlo S, Chen B, Hoover TR, Kondrashkina E, Nogales E, Nixon BT. The structural basis
for regulated assembly and function of the transcriptional activator NtrC. *Genes &*
*development* **20**, 1485-1495 (2006).
- 5. Batchelor JD, *et al.* Structure and regulatory mechanism of Aquifex aeolicus NtrC4:
variability and evolution in bacterial transcriptional regulation. *J Mol Biol* **384**, 1058-1075
(2008).
- 6. Howlett GJ, Minton AP, Rivas G. Analytical ultracentrifugation for the study of protein
association and assembly. *Curr Opin Chem Biol* **10**, 430-436 (2006).
